# Neuronal genes deregulated in Cornelia de Lange Syndrome respond to removal and re-expression of cohesin

Felix D. Weiss [1,8,11], Lesly Calderon[1,9,11], Yi-Fang Wang[2], Radina Georgieva [1,3], Ya Guo[1,10], Nevena Cvetesic [3], Maninder Kaur[4], Gopuraja Dharmalingam[2], Ian D. Krantz[4,5,6], Boris Lenhard [3,7], Amanda G. Fisher [1] & Matthias Merkenschlager [1✉]

Cornelia de Lange Syndrome (CdLS) is a human developmental disorder caused by mutations that compromise the function of cohesin, a major regulator of 3D genome organization. Cognitive impairment is a universal and as yet unexplained feature of CdLS. We characterize the transcriptional profile of cortical neurons from CdLS patients and find deregulation of hundreds of genes enriched for neuronal functions related to synaptic transmission, signalling processes, learning and behaviour. Inducible proteolytic cleavage of cohesin disrupts 3D genome organization and transcriptional control in post-mitotic cortical mouse neurons, demonstrating that cohesin is continuously required for neuronal gene expression. The genes affected by acute depletion of cohesin belong to similar gene ontology classes and show significant numerical overlap with genes deregulated in CdLS. Interestingly, reconstitution of cohesin function largely rescues altered gene expression, including the expression of genes deregulated in CdLS.

[1] Lymphocyte Development Group, Epigenetics Section, MRC London Institute of Medical Sciences, Institute of Clinical Sciences, Faculty of Medicine, Imperial College London, London, UK. [2] MRC London Institute of Medical Sciences, Institute of Clinical Sciences, Faculty of Medicine, Imperial College London, London, UK. [3] Computational Regulatory Genomics Group, Epigenetics Section, MRC London Institute of Medical Sciences, Institute of Clinical Sciences, Faculty of Medicine, Imperial College London, London, UK. [4] Division of Human Genetics, The Department of Pediatrics, The Children's Hospital of Philadelphia, Philadelphia, PA, USA. [5] The Perelman School of Medicine at The University of Pennsylvania, Philadelphia, PA, USA. [6] Department of Pathology and Laboratory Medicine, The Children's Hospital of Philadelphia, Philadelphia, PA, USA. [7] Sars International Centre for Marine Molecular Biology, University of Bergen, Bergen, Norway. [8] Present address: Institute of Innate Immunity, University of Bonn, Bonn, Germany. [9] Present address: Research Institute of Molecular Pathology, Vienna, Austria. [10] Present address: School of Life Sciences and Biotechnology, Shanghai Jiao Tong University, Shanghai, China. [11] These authors contributed equally: Felix D. Weiss, Lesly Calderon. ✉email: matthias.merkenschlager@lms.mrc.ac.uk

Three-dimensional (3D) genome organization into topologically associated domains (TADs), contact domains and chromatin loops spatially compartmentalises genes and enhancers and facilitates transcriptional control by gene regulatory elements[1–4]. 3D genome organization is achieved through the activity of architectural proteins, including the cohesin complex. Initially identified as essential for chromosomal integrity during the cell cycle, cohesin is now known to cooperate with the DNA binding protein CTCF in 3D chromatin contacts essential for transcriptional control[3–8]. Mechanistically, cohesin increases 3D contact probabilities of sequence elements, including enhancers and promoters, within boundaries marked by CTCF binding sites in convergent orientation[3,5,6]. In addition, a subset of promoters and enhancers are direct targets of CTCF, and genes that contact enhancers via CTCF-based cohesin loops are highly susceptible to deregulation when CTCF or cohesin are perturbed[3,9,10]. Mammalian genes have been classified into those that are controlled mainly by their promoters, and those that primarily depend on distal enhancers for transcriptional regulation[11]. This difference in regulatory 'logic' broadly separates ubiquitously expressed, promoter-centric 'housekeeping' genes from enhancer-controlled tissue-specific genes[12]. While the loss of cohesin affects the transcription of a limited number of genes, enhancer-associated[7,13] and inducible genes[14], including neuronal activity-dependent genes[15], are frequently deregulated when 3D organization is perturbed by the loss of cohesin.

Heterozygous or hypomorphic germline mutations of cohesin and associated factors such as the cohesin loading factor NIPBL result in a human developmental disorder known as Cornelia de Lange Syndrome (CdLS)[16–18]. All CdLS patients show a degree of intellectual disability, and 60–65% have autism spectrum disorder (ASD), often in the absence of structural brain abnormalities or neurodegeneration[17,19,20]. Experimental perturbations of neuronal cohesin or NIPBL during mouse development have shown changes in animal behaviour as well as abnormal neuronal morphology[21], migration[22] and gene expression[21–24].

NIPBL has historically been considered as separate from the core cohesin complex, and with distinct[22,25–27] and sometimes even antagonistic activities[28]. However, current structural[29] and functional[30] evidence indicate that NIPBL is integral to the cohesin complex in its loading state[29] and contributes to ATPase activity[30] during loop extrusion, the process thought to form chromatin loops, TADs and contact domains[5]. Accordingly, the depletion of NIPBL and cohesin have similar or identical effects on 3D genome organization[7,8]. Consistent with the function of NIPBL as a loading factor for cohesin, CdLS patient-derived $NIPBL^{+/-}$ lymphoblastoid cells[31], $Nipbl^{+/-}$ mouse embryonic fibroblasts[32] and fetal liver cells[33] show reduced global or local cohesin binding[31–33] and defective 3D chromatin contacts[32,33]. Deregulated genes show reduced cohesin binding in $Nipbl^{+/-}$ embryonic mouse brain[34]. Nevertheless, it remains unclear to what extent gene expression changes in cells with $Nipbl$ mutations relate to cohesin or potential additional functions of NIPBL (ref. [22,25–27]).

Although aberrant gene expression is a likely cause of neuronal dysfunction in CdLS, studies of gene expression in CdLS patient cells have to date been limited to induced pluripotent stem cells (iPSCs)[35], cardiac[35] and lymphoblastoid cells[31]. We, therefore, do not know whether CdLS neurons show aberrant gene expression, and how gene expression in CdLS relates to perturbations of neuronal gene expression in response to cohesin deficiency. In this study, we examine gene expression in neuronal nuclei isolated from post-mortem cerebral cortex of CdLS patients. We discover prominent downregulation of hundreds of genes enriched for important neuronal functions, including synaptic transmission, signalling processes, learning and behaviour, and neuroprotective sphingolipid metabolism. Deregulated genes show significant overlap with ASD. These findings support the idea that altered gene expression may contribute to neuronal dysfunction in CdLS. We establish experimental models that allow for the inducible degradation of the cohesin subunit RAD21 in post-mitotic primary cortical mouse neurons. These reveal that cohesin is continuously required to sustain neuronal gene expression. A significant number of cohesin-dependent genes are also deregulated in CdLS. Importantly, the expression of these genes is rescued by reconstitution of functional cohesin, indicating that at least some of these changes may be reversible.

## Results

**Transcriptomic characterization of CdLS patient neurons.** We sourced frozen post-mortem prefrontal cortex from four CdLS patients aged 19–48 years and six age-matched controls (Supplementary Fig. 1a). Three of the patients had heterozygous mutations in *NIPBL*, the gene most frequently mutated in CdLS (ref.[17,36], Fig. 1a, Supplementary Fig. 1a, b). No mutations in *NIPBL* or other CdLS genes such as *RAD21, SMC1A* or *HDAC8* were found in the fourth patient, consistent with the lack of identifiable mutations in ~30% of CdLS patients[36].

To characterise gene expression in CdLS patient neurons we isolated nuclei from the prefrontal cortex, stained with the neuronal marker NeuN (ref.[37]), and sorted NeuN positive and NeuN negative nuclei by flow cytometry (Fig. 1b, Supplementary Fig. 1c). We generated ATAC-seq, CAGE and RNA-seq data from neuronal nuclei. ATAC-seq and CAGE signals at enhancers did not show CdLS-specific features due to extensive inter-individual variation. However, CAGE analysis of promoters showed separation of CdLS patient and control samples along the major principle component PC1, and revealed significant deregulation of 766 gene promoters in CdLS neurons (adj $P < 0.05$; 358 upregulated, 408 downregulated; Supplementary Fig. 2a–c, Supplementary Data 1). Gene promoters that were downregulated in CdLS neurons were enriched for the gene ontology (GO) terms synaptic signalling and organization, ion and neurotransmitter transport, axon development, behavior and cognition, while upregulated promoters showed no striking functional enrichment (Supplementary Fig. 2d, Supplementary Data 2). RNA-seq identified 617 differentially expressed genes (adj $P < 0.05$) in CdLS neurons, 310 genes were up- and 307 were downregulated (Fig. 1c, Supplementary Fig. 3a). In close agreement with CAGE, genes that were downregulated by RNA-seq were related to synaptic transmission, signalling processes, learning and behaviour, and sphingolipid metabolism, while upregulated genes showed no striking functional enrichment (Fig. 1d, Supplementary Data 4).

RNA-seq and CAGE of CdLS neurons showed highly significant overlap (Odds ratio = 11.86, $P < 2.2e-16$). The direction of gene deregulation by RNA-seq and CAGE was highly correlated ($R = 0.68$, $P = < 2.2e-16$ for genes found deregulated by RNA-seq or CAGE, and $R = 0.95$, $P = < 2.2e-16$ for genes found deregulated by both RNA-seq and CAGE; Supplementary Fig. 2e).

A majority of CdLS patients have ASD in addition to intellectual disability[17,20]. Clinical records relating to the brain samples examined here indicate that all four patients had cognitive impairment and three had confirmed ASD features (CDL380P, CDL744P, CDL764P, there was no information on ASD status for 2082). We, therefore, asked whether genes implicated in ASD were deregulated in CdLS. Genes found deregulated in the prefrontal cortex of patients with idiopathic ASD (ref.[38]) showed strong overlap with genes deregulated in CdLS (Odds ratio = 6.25, $P < 2.2e-16$). The direction of gene expression changes in CdLS and idiopathic ASD was highly correlated ($R = 0.56$, $P < 2.2e-16$ for genes deregulated in idiopathic ASD, $R = 0.85$, $P < 2.2e-16$ for

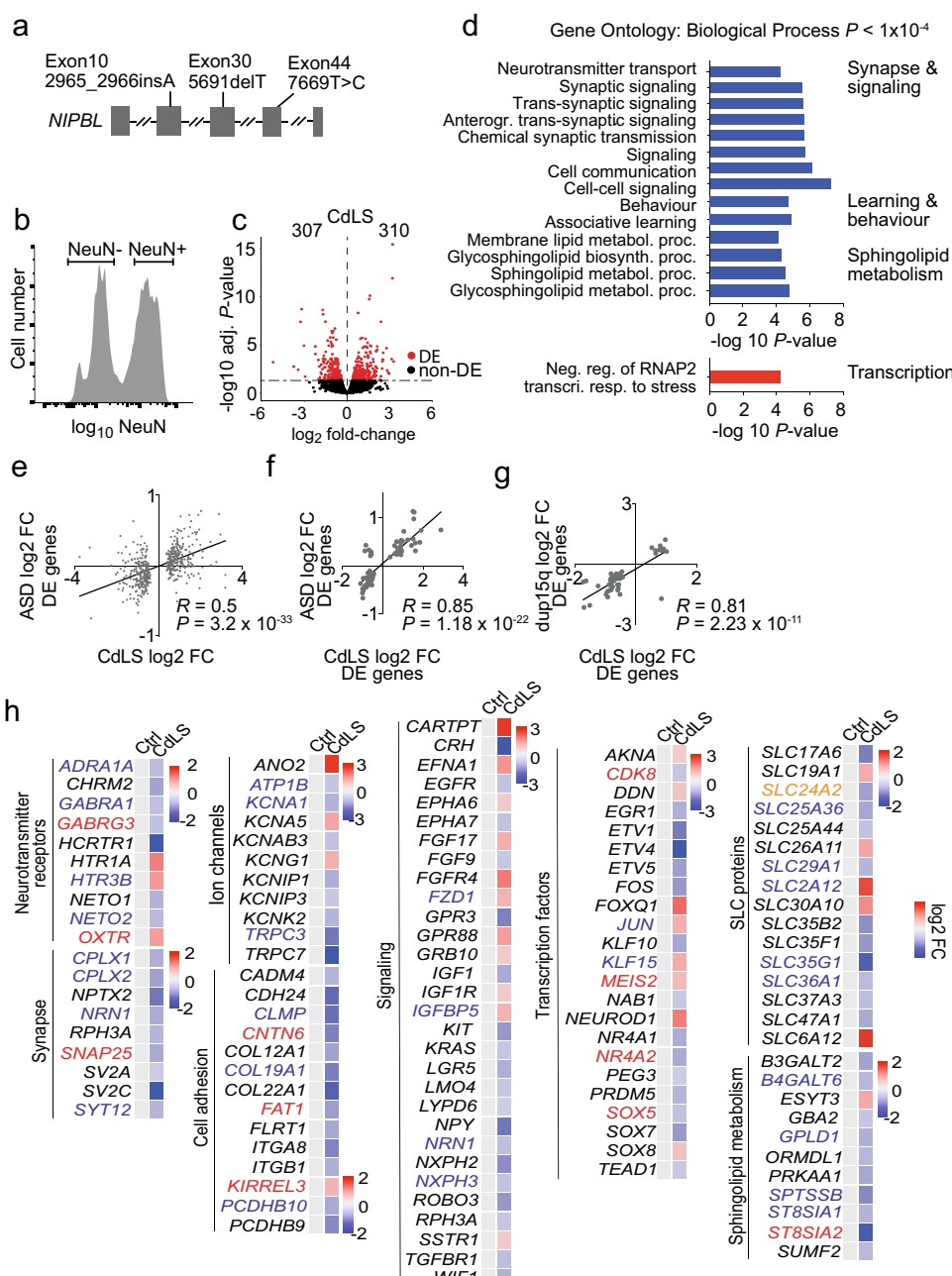

**Fig. 1 CdLS patient brains display abnormal neuronal gene expression. a** *NIPBL* mutations in CdLS patients. **b** Flow cytometry histogram of NeuN-stained nuclei from post-mortem tissue. Sort gates for NeuN-positive and -negative nuclei are indicated. **c** Volcano plot of gene expression fold-change versus adjusted *P* value of NeuN-positive nuclei from CdLS patients ($n = 4$). 307 genes were down- and 310 genes were upregulated (RUVg k = 2, adj. $P < 0.05$, Wald Test, Benjamini-Hochberg adjusted). Differentially expressed (DE) genes are shown in red. **d** Bar graph of individual GO terms and broad categories ($P < 1 \times 10^{-4}$, Wallenius approximation). Terms represented by downregulated genes are shown in blue, those by upregulated genes in red. **e** Scatter plot of gene expression comparing $\log_2$ fold-change of differentially expressed genes (adj $P < 0.05$) in idiopathic ASD, and the same genes in NeuN-positive CdLS samples. ($R$ = Pearson correlation coefficient; $P = 3.2e-33$, two-sided $F$ test). **f** Scatter plot of $\log_2$ fold-change of shared deregulated genes (adj $P < 0.05$) in both ASD and NeuN-positive CdLS samples. ($R$ = Pearson correlation coefficient; $P = 1.18e-22$, two-sided $F$ test). **g** Scatter plot of $\log_2$ fold-change of genes that were deregulated in both duplication of chromosome 15q.11.2-13.1 (dup15q) and NeuN-positive CdLS samples. ($R$ = Pearson correlation coefficient; $P = 2.23e-11$, two-sided $F$ test). **h** Heatmap of $\log_2$ fold-changes for deregulated genes in NeuN-positive CdLS samples. Genes are grouped by functional categories. Genes highlighted in red are in the SFARI database[41], those in blue were deregulated in ASD and/or dup15q, genes highlighted in orange meet both criteria.

genes deregulated in both idiopathic ASD and CdLS; Fig. 1e, f). Strong overlap of deregulated genes and correlated changes in the direction of gene expression were also seen between CdLS and ASD caused by the duplication of chromosome 15q.11.2-13.1 (ref. [35], $R = 0.85$, $P < 2.2e-16$; Fig. 1g). These data reveal similarities between the transcriptional programs of neurons in CdLS and in idiopathic and syndromic ASD.

Deregulated genes in CdLS neurons included transcription factors, such as the ASD risk gene *SOX5*, neurotransmitter receptors and associated proteins for glutamatergic (*NETO1*,

NETO2), GABAergic (*GABRA1*, *GABRG3*) and serotonergic (*HTR1A*, *HTR3B*) signalling, as well as potassium (*KCNA1*, *KCNG1*, *KCNK1*), sodium (*ATP1B*), and chloride (*ANO2*) channels that control electrochemical gradients and neuronal transmission. Components of the Wnt (*FZD1*), Ephrin (*EFNA1*, *EPHA6/7*), EGF (*EGFR*), FGF (*FGF9/17*, *FGFR4*) and TGF-beta (*TGFBR1*) signalling pathways were deregulated, as were genes involved in neuronal adhesion including protocadherins (*PCDHB9/10*), which had previously been shown to depend on cohesin and CTCF for their correct expression[39], and enzymes involved in the synthesis of sphingolipids with neuroprotective functions (*B3GALT2*, *ST8SIA1/2*, Fig. 1h).

RNA-seq of NeuN-negative nuclei showed upregulation of immune response genes that were not seen in NeuN-positive nuclei, including interferon response, JAK-STAT signalling, and the complement pathway (Supplementary Fig. 4, Supplementary Data 5, 6). This suggests that the deregulation of neuronal genes in CdLS is accompanied by the activation of inflammatory genes in non-neuronal cells. There was significant overlap between changes in gene expression in human CdLS with published[23] gene expression in *Nipbl* heterozygous mouse brain at embryonic day 13.5 (e13.5; *Nipbl*$^{+/-}$ whole brain versus human CdLS NeuN positive and NeuN negative nuclei combined; $P = 1.32e-14$, Odds ratio = 2.05) and with gene expression in *Nipbl*$^{+/-}$ cortical neurons in explant culture (*Nipbl*$^{+/-}$ cortical neurons versus human CdLS NeuN positive neurons; Supplementary Data 7; $P = 8.38e-06$, Odds ratio = 3.03).

In summary, these data demonstrate that neurons from CdLS patients show deregulated expression of genes that have important neuronal functions, consistent with neuronal dysfunction in CdLS. They raise the question of whether deregulated gene expression in CdLS is related to reduced cohesin function and—if so—whether the deregulation of neuronal genes is a direct consequence of reduced cohesin function in post-mitotic neurons or secondary to a role for cohesin in neural development.

**Temporal control over cohesin levels in neurons**. Addressing the question of whether cohesin is directly required for neuronal gene expression requires experimental systems that enable acute cohesin withdrawal from post-mitotic neurons, ideally in a reversible manner. To control the expression of RAD21, a cohesin subunit associated with CdLS (ref. [16,17]), we utilized *Rad21*$^{Tev}$, an allele that encodes RAD21 protein cleavable by tobacco etch virus (TEV) protease[14,40]. *Rad21*$^{Tev/Tev}$ neurons express endogenous RAD21-TEV as their sole source of RAD21. We established explant cultures from *Rad21*$^{Tev/Tev}$ cortices at e14.5 under conditions that enrich for post-mitotic neurons and promote neuronal maturation (Fig. 2a, Supplementary Fig. 5a, b).

We used lentivirus to transduce *Rad21*$^{Tev/Tev}$ neurons with TEV protease fused to a tamoxifen-responsive estrogen receptor hormone binding domain (ERt2-TEV). Constructs were tagged by V5 and t2a-fused GFP (ERt2-TEV) for identification of ERt2-TEV-expressing cells (Fig. 2b, Supplementary Fig. 5c). ERt2-TEV remained cytoplasmic until nuclear translocation of ERt2-TEV was triggered, which occurred within minutes of 4-hydroxy tamoxifen (4-OHT) addition (Fig. 2b, Supplementary Fig. 5c). We monitored the cleavage of RAD21-TEV by western blotting and found that RAD21-TEV expression was reduced to ~15% of pre-treatment levels within 8 h of 4-OHT addition, and remained at this level for at least 24 h (Fig. 2c). In a second approach, we transduced *Rad21*$^{Tev/Tev}$ neurons with a lentiviral construct that constitutively expresses Tet-On advanced transactivator (rtTA) and RFP. Addition of doxycycline leads to the expression of TEV protease with an exogenous nuclear localization sequence (NLS-TEV, Supplementary Fig. 6a, b). Induction of NLS-TEV depleted

~70% of RAD21-TEV within 24 h of doxycycline addition (Supplementary Fig. 6c). These systems establish temporal control over cohesin levels in primary post-mitotic neurons.

**Cohesin is continuously required for neuronal gene expression**. We utilized temporal control of RAD21-TEV cleavage to quantify the impact of acute cohesin depletion on gene expression in post-mitotic neurons. RNA-seq 24 h after 4-OHT-induced nuclear translocation of ERt2-TEV identified 750 deregulated (adj $P < 0.05$) genes, of which 463 were down- and 287 were upregulated (Fig. 2d, Supplementary Data 8). GO term analysis showed that downregulated genes were enriched for biological processes similar to those observed for downregulated genes in CdLS, in particular synapse and signalling, cell adhesion, neuron development and ion transport. As observed in CdLS neurons, no striking enrichment was observed for upregulated genes (Fig. 2e, Supplementary Data 9). Genes deregulated by ERt2-TEV-mediated RAD21-TEV cleavage showed significant overlap with SFARI ASD risk genes[41] (Odds ratio = 2.19, $P = 1.16e-08$).

RNA-seq 24 h after doxycycline-dependent NLS-TEV induction in *Rad21*$^{Tev/Tev}$ neurons identified 570 down- and 150 upregulated genes (adj $P < 0.05$). As observed for the nuclear translocation of ERt2-TEV, downregulated genes were enriched for adhesion, signalling, synaptic function and development (Supplementary Fig. 5d, e, Supplementary Data 10, 11). There was strong overlap between genes deregulated in both systems (Odds ratio = 20.35, $P < 2.2e-16$), especially for downregulated genes (Odds ratio = 53.22, $P < 2.2e-16$, Fig. 2f), and the direction of gene deregulation was highly concordant ($R = 0.75$, $P < 2.2e-16$, Fig. 2g). There was extensive overlap between genes deregulated by acute cohesin depletion with genes deregulated in *Nipbl*$^{+/-}$ neurons (this study) and embryonic *Nipbl*$^{+/-}$ brain[23], but less so with genes deregulated in other mouse models of neuronal dysfunction[42–46] (Supplementary Fig. 7).

Reduced mRNA expression translated into reduced levels of protein in RAD21-TEV neurons, as illustrated for NLGN1, encoded by *Nlgn1* (Supplementary Fig. 8, Supplementary Data 12). Hence, acute depletion of RAD21-TEV in post-mitotic neurons established that cohesin is continuously required to sustain the expression of genes that mediate important neuronal functions.

The *Pcdhb* cluster is a classic example of CTCF-based cohesin-mediated enhancer-promoter connections[3,39], and 15 of 17 expressed *Pcdhb* genes were deregulated by inducible RAD21-TEV cleavage. Circular chromosomal conformation capture and sequencing (4C-seq) using the *Pcdhb* enhancers HS18-20 as the viewpoint showed that RAD21-TEV cleavage disrupted 3D contacts, and decreased interactions between *Pcdhb* promoters and enhancers (Fig. 2h). Genes that were downregulated in response to acute cohesin depletion in primary neurons were significantly enriched for the binding of RAD21 (Fig. 2i, Odds Ratio = 2.66, $P = 2.32e-13$) and CTCF (Fig. 2i, Odds Ratio = 2.38, $P = 3.39e-11$). Our finding that cohesin deficiency predominantly affects genes related to specialized neuronal functions is consistent with models that distinguish ubiquitously expressed, promoter-centric 'housekeeping' genes from enhancer-controlled tissue-specific genes[11,12], and with previous observations that enhancer-associated genes are preferentially deregulated by depletion of cohesin[7,13]. Accordingly, genes that were downregulated in response to acute cohesin depletion in primary neurons were significantly enriched for proximity to enhancers (Fig. 2i, Odds ratio = 3.78, $P < 2.2e-16$). These data suggest a mechanism by which RAD21-TEV cleavage disrupts cohesin-mediated chromatin contacts, including enhancer-promoter interactions.

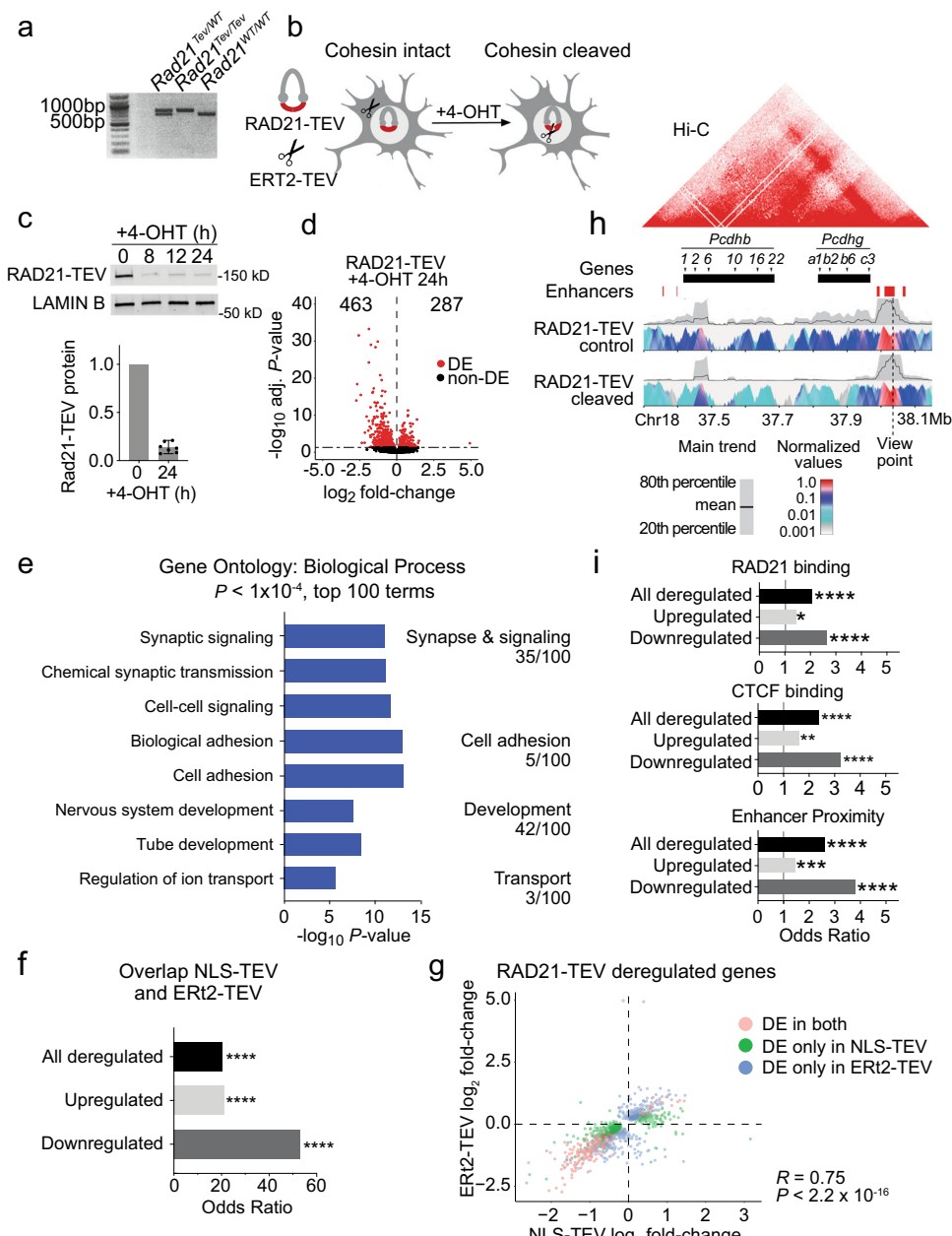

**Fig. 2 Cohesin is continuously required to sustain neuronal gene expression. a** PCR analysis of *Rad21* alleles from *Rad21*<sup>*Tev/WT*</sup>, *Rad21*<sup>*Tev/Tev*</sup> and *Rad21*<sup>*WT/WT*</sup> mice. **b** Schematic of ERt2-TEV-dependent RAD21-TEV degradation. **c** Western blot of RAD21-TEV protein expression over a time course of 4-OHT treatment. Bar plot of RAD21-TEV protein expression normalized to LAMIN B ($n = 7$, h = hours). **d** Volcano plot of gene expression fold-change versus adjusted *P* value in RNA-seq of RAD21-TEV neurons transduced with ERt2-TEV and treated with 4-OHT for 24 h ($n = 3$). A total of 463 genes were down- and 287 genes were upregulated (adj $P < 0.05$, Wald Test, Benjamini-Hochberg adjusted). Differentially expressed (DE) genes are shown in red. **e** Bar graph of individual GO terms and broad categories ($P < 1 \times 10^{-4}$, Wallenius approximation) in RNA-seq of RAD21-TEV neurons treated as in (**d**). Terms represented by downregulated genes are shown in blue. Upregulated genes showed no GO term enrichment at $P < 1 \times 10^{-4}$. **f** Enrichment of shared deregulated genes (adj $P < 0.05$) in response to acute cohesin depletion induced by ERt2-TEV and NLS-TEV. One-sided Fisher's exact test was applied for the odds ratio and *P* value. All comparisons $P < 2.22e\text{-}16$. **g** Scatter plot of gene expression, comparing log₂ fold-change of deregulated genes (adj $P < 0.05$) in response to acute cohesin depletion induced by ERt2-TEV and NLS-TEV (*DE* differentially expressed, *R* = Pearson correlation coefficient; $P < 2.2e\text{-}16$, two-sided *F* test). **h** 4 C analysis of chromatin interactions at the *Pcdhb* locus. Top: Hi-C representation of domain structure at the *Pcdhb* locus in mouse cortical neurons[95], with selected genes and enhancers. Bottom: contact profiles of enhancers HS18-20 in control cells (top panel) and RAD21-TEV cleaved cells (bottom panel). A dashed line indicates the enhancer site and viewpoint. A grey band displays the 20th to 80th percentiles and the black line within shows mean values for 40 kb windows. The colour panel shows the mean contact intensities for multiple window sizes from 2 kb to 5 kb for $n = 4$ independent biological replicates. All replicates showed comparable results and were merged to generate this figure. **i** Genes significantly deregulated (adj $P < 0.05$) by ERt2-TEV mediated RAD21-TEV degradation are enriched for the binding of cohesin (top) CTCF (center) and proximity to neuronal enhancers (bottom, see methods). One sided Fisher's exact test was applied for the odds ratio and *P* value. *$P < 0.05$, **$P < 0.01$, ***$P < 0.001$, ****$P < 0.0001$.

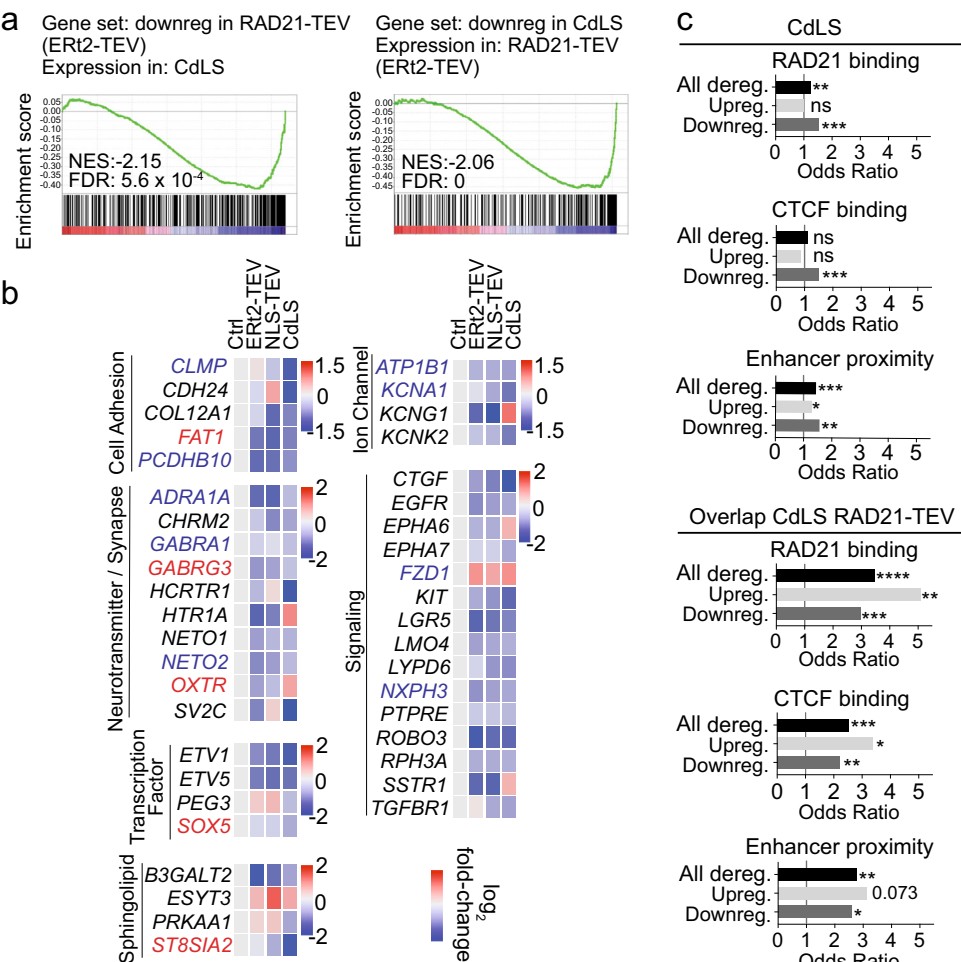

**Fig. 3 Cohesin-dependent gene regulation in CdLS neurons. a** GSEA of RAD21-TEV downregulated genes in CdLS NeuN-positive RNAseq (left). GSEA of CdLS NeuN-positive downregulated genes in RAD21-TEV (right). *NES* normalized enrichment score, *FDR* false discovery rate. **b** Heatmap of log₂ fold-change for selected genes from CdLS NeuN-positive and RAD21-TEV samples separated into broad functional categories. Genes shown were significantly deregulated in CdLS and their mouse orthologs deregulated in at least one RAD21-TEV (ERt2-TEV or NLS-TEV) system. Genes highlighted in red were identified in the SFARI database, those in blue were deregulated in ASD, genes highlighted in orange fulfil both criteria. **c** Comparison of RAD21 (cohesin) binding (top), CTCF binding (center) and enhancer proximity (bottom, see methods) of genes deregulated (adj *P* < 0.05) in NeuN-positive CdLS (left) compared to genes deregulated in both NeuN-positive CdLS and RAD21-TEV (right). Up- and downregulation indicates the direction of deregulation in CdLS NeuN-positive RNAseq. One sided Fisher's exact test was applied for the odds ratio and *P* value. *P < 0.05, **P < 0.01, ***P < 0.001, ****P < 0.0001.

**Cohesin-dependent gene regulation and CdLS**. We next compared cohesin-dependent genes identified in $Rad21^{Tev/Tev}$ neurons with gene expression in CdLS patient neurons. Gene set enrichment analysis (GSEA) showed remarkable concordance between genes that were downregulated in CdLS and in mouse neurons acutely depleted of RAD21-TEV. Genes that were downregulated in $Rad21^{Tev/Tev}$ neurons 24 h after ERt2-TEV-induced cohesin depletion showed, as a set, significant downregulation also in CdLS patient neurons (NES = −2.15, FDR = 5.6 × 10⁻⁴, Fig. 3a). The reciprocal test confirmed this concordance and showed that genes downregulated in CdLS were overall downregulated in $Rad21^{Tev/Tev}$ neurons (NES = −2.06, FDR = 0.00, Fig. 3a). For upregulated genes there was no such concordance (NES = 1.28, FDR = 0.13 and NES = +1.04, FDR = 0.35, respectively).

Accordingly, a significant number of genes were deregulated both in CdLS patient neurons and after acute cohesin depletion of mouse neurons (51 in CdLS and ERt2-TEV, 57 in CdLS and NLS-TEV, 81 shared overall, Odds ratio = 2.48, P = 4.87e-11, of which 55 were downregulated, Odds ratio = 3.14, P = 4.06e-11). The majority of shared genes were deregulated in the same direction

(40 of 51 for ERt2-TEV, 42 of 57 for NLS-TEV, Fig. 3b). Of the 81 genes that overlapped between CdLS and one or both of the RAD21-TEV depletion systems, 23 were ASD risk genes[41] or deregulated in ASD (Ref. [38]). These shared ASD genes mediate important neuronal functions in cell adhesion (*CLMP, FAT1, PCDHB10*) signalling (*PRKD1, NXPH3, FZD1, PHLDA1, PMEPA1, CAMK2G*), ion channels (*ATP1B1, KCNA1*), synaptic transmission (*LGI2, GABRA1, GABRG3, NETO2*), transcription (*SOX5, CECR2, CELF6*) and sphingolipid metabolism (*ST8SIA2*) (Fig. 3b, Supplementary Table 1).

There was slight, but statistically significant enrichment for binding of RAD21 and CTCF to genes that were downregulated in CdLS neurons (Fig. 3c). This enrichment increased considerably for genes that were deregulated both in CdLS and by the inducible cleavage of RAD21-TEV (compare Odds ratios in Fig. 3c). Genes that were deregulated both in CdLS and in acutely cohesin-depleted $Rad21^{Tev/Tev}$ neurons showed significant enrichment for proximity to human neuronal enhancers[47] (Fig. 3c).

Gene expression changes in Rett syndrome, but not other human neurological diseases[43,48–50], showed significant overlap

with $Rad21^{Tev/Tev}$ neurons (Odds ratio = 2.06, $P$ = 4.64e-10; Supplementary Fig. 9).

In summary, a subset of genes deregulated in CdLS neurons bore hallmarks of cohesin target genes, and these genes were enriched in the overlap between CdLS and cohesin-dependent genes, as identified by acute depletion of cohesin in $Rad21^{Tev/Tev}$ neurons.

**Rescue of cohesin-dependent gene expression**. Postnatal intervention can alleviate transcriptional changes and abnormal behaviour in mouse models of Rett Syndrome[51,52]. In light of this, we asked whether gene expression changes induced by acute cohesin depletion in $Rad21^{Tev/Tev}$ neurons can be reversed when cohesin levels are restored. Doxycycline-induced NLS-TEV expression was reversible, and transient RAD21-TEV cleavage was followed by a return of RAD21-TEV protein to control levels after removal of doxycycline (Fig. 4a, b). Remarkably, comparison of RNA-seq during RAD21-TEV depletion (24 h) and after RAD21-TEV rescue (day 7) showed that the vast majority of genes that were found deregulated after 24 h (695 of 720 or 96.5%) were fully rescued upon restoration of cohesin levels (Fig. 4c, e; Supplementary Data 13). A small number of residual genes were refractory to rescue and remained deregulated. These included neuronal signalling (*Gfra1, Spon1, Scg2, Cntnap3*), transcription (*Mycl1, Cited2, Maml3, Lzts1*) and splicing factors (*Rbfox1*). To ask whether the observed rescue of gene expression reflected adaptation of neurons to reduced cohesin function we carried out RNA-seq experiments after prolonged depletion of RAD21-TEV. These experiments confirmed that rescue of gene expression required the restoration of RAD21 expression and was not due to the adaptation of neurons to reduced cohesin expression (Supplementary Fig 10; Supplementary Data 14).

Interestingly, all 57 genes that were deregulated both in CdLS and in response to NLS-TEV-mediated cohesin depletion were rescued by restoring cohesin expression. These included 14 genes implicated in ASD and shared between CdLS and NLS-TEV (Supplementary Table 1). Rescue was near-complete, regardless of the direction and the degree of the initial deregulation (Fig. 4d).

We identified 62 genes that were initially unaffected by acute cohesin depletion at 24 h, but were deregulated 7 days later, after cohesin levels had been restored to control levels (Fig. 4e). These de novo deregulated genes were associated with the development and cell communication and were highly enriched for genes that change expression during the maturation of control neurons in explant culture (day 10–17, adj $P$ < 0.05). Of 62 de novo deregulated genes, 46 were maturation genes (Odds ratio = 6.69, $P$ = 1.14e-12). Strikingly, the direction of deregulation of these genes was highly correlated with their regulation during neuronal maturation: genes that were upregulated during neuronal maturation in explant culture were also preferentially upregulated as a consequence of transient cohesin depletion ($P$ = 0.0134, FDR = 0.0134), while genes that were downregulated during neuronal maturation in explant culture were also preferentially downregulated as a consequence of transient cohesin depletion ($P$ = 1.51e-04, FDR = 3.02e-04) (Fig. 4f, g). Hence, while the majority of gene expression changes caused by loss of cohesin can be rescued by restoring cohesin levels, including the expression of genes deregulated in CdLS, the de novo deregulation of neuronal maturation genes illustrates that even transient cohesin depletion may have potentially damaging long-term secondary effects on neuronal gene expression.

## Discussion

We characterized gene expression in primary cortical neurons from CdLS patients and found deregulation of hundreds of genes enriched for important neuronal functions, including synaptic transmission, signalling processes, learning and behaviour and sphingolipid metabolism. These findings provide experimental support for suggestions that CdLS pathologies, including intellectual disability, may be mediated at least in part by deregulated gene expression. The transcriptomic profile of CdLS showed extensive similarities to ASD, both in terms of shared deregulated genes and in the direction of deregulation. This similarity is of particular interest given the high prevalence of ASD in CdLS patients.

To explore the role of cohesin in neuronal gene expression we established in vitro models for the acute depletion of cohesin in primary cortical mouse neurons. Proteolytic cleavage of the essential cohesin subunit RAD21 disrupted 3D organization and perturbed the expression of neuronal genes. These experiments establish that cohesin is continuously required to sustain neuronal gene expression.

We found significant concordance between cohesin-dependent genes in mouse neurons and the neuronal transcriptome in human CdLS. Genes deregulated both in cohesin-depleted mouse neurons and human CdLS neurons were enriched for cohesin binding and enhancer proximity. Tissue-specific genes primarily depend on distal enhancers for transcriptional regulation[11,12], and enhancer-associated genes are preferentially deregulated when 3D organization is perturbed by the loss of cohesin[7,13]. Our finding that cohesin deficiency predominantly affects enhancer-associated genes related to specialized neuronal functions is consistent with these models. Taken together with previous data demonstrating the impact of *Nipbl* heterozygosity on cohesin binding and 3D chromatin contacts[31–34], our analysis supports the idea that deregulation both in CdLS in vivo and in in vitro models for acute cohesin cleavage identifies a core set of genes that are enriched for binding of CTCF and cohesin, and that show enhancer-dependent tissue-specific expression. These genes are susceptible to deregulation because they rely on 3D contacts with distal regulatory elements, and their 3D contacts are directly dependent on cohesin. These genes related to signaling, neurotransmitter and synapse components, ion channels, and transcription factors. Interestingly, the great majority of gene expression changes precipitated by acute cohesin depletion were reversible upon restoration of cohesin function, including all 57 genes deregulated in both CdLS and NLS-TEV. Hence, cohesin rescues neuronal genes that are deregulated in CdLS.

In addition to directly cohesin-dependent genes, CdLS neurons also showed deregulation of genes that lacked hallmarks of direct cohesin targets and were not sensitive to acute cohesin depletion in mouse neurons. This may be due to cohesin-independent functions of NIPBL (ref. [22,25–27]) or to secondary effects of reduced cohesin function on the expression of other genes. Our data point to two mechanisms that may contribute to the indirect deregulation of genes in cohesin-deficient neurons. First, numerous genes related to neuronal maturation were deregulated late in response to acute cohesin depletion and remained deregulated after cohesin levels were restored. These genes showed a striking pattern of overshoot regulation, where genes upregulated during the maturation of control neurons were even more highly expressed after transient cohesin depletion and vice versa. Explant cultures contain >90% neurons, and the deregulation of these genes may therefore be driven by factors within or interactions between neurons. Second, analysis of NeuN negative nuclei from CdLS cortices showed highly significant upregulation of inflammatory genes, consistent with potentially damaging inflammatory responses by glial cells[53]. These data identify an inflammatory component to disrupted gene expression in CdLS, which may contribute to neuronal dysfunction via glia-neuron interactions. Interestingly, one of the gene ontologies

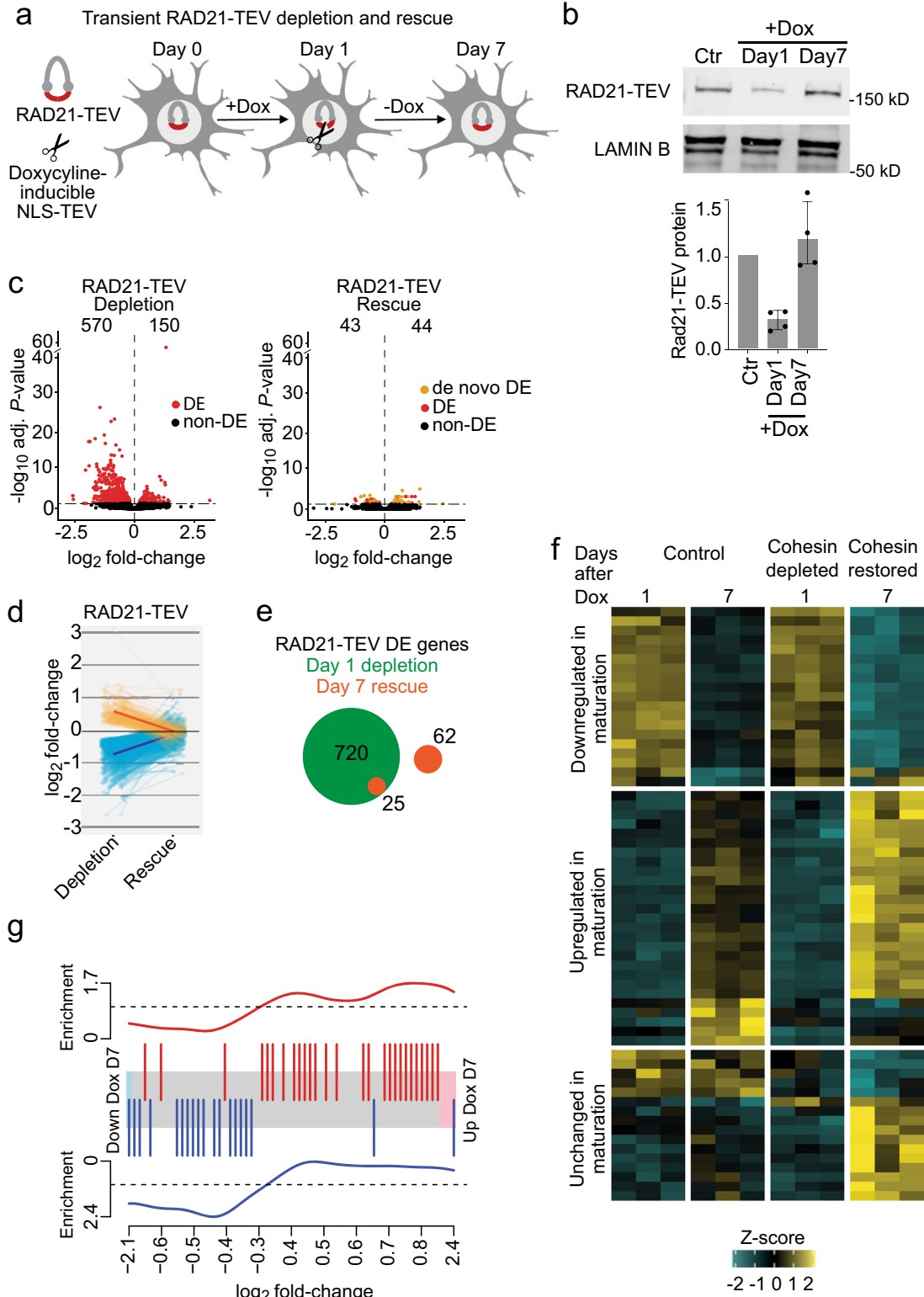

found deregulated in CdLS neurons, sphingolipid metabolism, is linked to neuroinflammation[54].

Our data raise the possibility that aspects of the CdLS phenotype may be reversible by postnatal intervention. Precedent for such rescue comes from pioneering studies in mouse models of Rett syndrome, where postnatal correction of *Mecp2* deficiency ameliorates deregulated gene expression and behavioural defects[51,52]. As the Rett protein MECP2, cohesin exerts its effects on transcription by interfacing with chromatin marking systems[55], and the dynamics of cohesin on chromatin is controlled in part by acetylation and deacetylation of its SMC subunits[18]. The acetylation/deacetylation cycle is a prime target for pharmacological intervention, as are bromodomain proteins such as BRD4, which connect acetylation to transcription[56]. Mutations in the histone deacetylase HDAC8 and the bromodomain protein BRD4 can cause CdLS-like disease[18,57]. Epigenetic drugs[56,58] may therefore

**Fig. 4 Rescue of cohesin-dependent gene expression. a** Schematic of NLS-TEV mediated RAD21-TEV degradation and rescue. **b** Western blot of RAD21-TEV protein expression over a time course of Dox treatment (6 h, 100 ng/ml). Bar plot of RAD21-TEV protein expression normalized to LAMIN B ($n = 4$). **c** Volcano plots of gene expression $\log_2$ fold-change versus adjusted $P$ value in RAD21-TEV + Dox treated neurons before and after rescue of RAD21-TEV expression ($n = 3$). Left: 570 genes were down- and 150 genes were upregulated following RAD21-TEV depletion (adj $P < 0.05$, Wald Test, Benjamini-Hochberg adjusted). Right: 43 genes were down- and 44 genes were upregulated after rescue of RAD21-TEV expression (adj $P < 0.05$, shown in red and orange, Wald Test, Benjamini-Hochberg adjusted). Genes in red are significantly deregulated in both RAD21-TEV depletion and rescue, genes in orange are significantly deregulated only after rescue (DE differentially expressed). **d** $\log_2$ fold-change of significantly deregulated genes (adj $P < 0.05$) following RAD21-TEV depletion (left) and after rescue (right). **e** The number of significantly deregulated genes (adj $P < 0.05$) after RAD21-TEV depletion (in green) and after rescue of RAD21-TEV expression (in orange). DE differentially expressed. **f** Heatmap of de novo deregulated genes following RAD21-TEV rescue (adj $P < 0.05$) and their maturation trajectory in control (left) and +Dox treated samples (right). **g** Barcode plot of de novo deregulated genes following RAD21-TEV rescue (adj $P < 0.05$) and their enrichment for directionality in maturation. Top: significantly de novo upregulated genes following rescue are enriched for genes upregulated during neuronal maturation. Bottom: de novo downregulated genes after cohesin rescue are enriched for genes that are downregulated during neuronal maturation.

provide alternatives to gene replacement approaches as potential therapeutics in CdLS.

## Methods

**Mice**. Mouse work was done under a project licence issued by the UK Home Office, UK following review by the Imperial College London Animal Welfare and Ethical Review Body (AWERB). Mice were maintained under SPF conditions with food and water ad libitum and a 12 h light/dark cycle. For timed pregnancies, the day of the vaginal plug was counted as day 0.5. $Rad21^{Tev/Tev}$ mice[14,40] on a mixed C57BL/6 129 background and $Nipbl^{+/-}$ mice[23] on a CD1 background have been described.

**Neuronal cultures**. Mouse cortices were dissected and dissociated on as described[59] on e14.5 for $Rad21^{Tev/Tev}$ and e17.5-e18.5 for $Nipbl^{+/-}$. Cells were plated at $1.25 \times 10^5$ cells/cm$^2$ in wells coated overnight with 0.1 mg/ml poly-D-lysine (Millipore) and one-half of the media in each well was replaced every 3 days. Cultures were treated with 5 μM Cytosine β-D-arabinofuranoside (Ara-C, Sigma) at day 2-4 for $Nipbl^{+/-}$ and day 5 for $Rad21^{Tev/Tev}$. $Nipbl^{+/-}$ neurons were used for RNA-seq on day 10. Neurons were plated on 12 mm coverslips coated with poly-D-lysine (VWR) for immunofluorescence staining.

**Lentivirus construction and packaging**. Cloning was performed by PCR amplification of cDNA using Phusion taq (Invitrogen) or Q5 polymerase (NEB), restriction digestion, gel purification of fragments and ligation into the target vector. To generate the ERt2-TEV lentiviral construct, Cas9 was removed from the lentiviral plasmid FUG-T2A-CAS9, which was a gift from Dr. Bryan Luikart (Addgene plasmid #75346)[60], by restriction enzyme digestion and replaced with v5-ERt2-TEV. To generate the NLS-TEV lentiviral construct, Gibson assembly was performed using Lenti-iCas9-Neo backbone and an NLS-TEV construct kindly provided by Dr. Kikue Tachibana. Lentivirus was generated as described previously[61] with minor modifications. Briefly, 293 T cells were co-transfected with the expression vector, together with packaging plasmids pCMV-VSV-G and pCMV-delta8.9 using PEI (Polysciences Inc.) 48 h after transfection, supernatant containing viral particles was collected and concentrated by ultracentrifugation. The titer of lentivirus was measured by transducing $5 \times 10^4$ 293 T cells per well of a 24-well plate at serial dilutions, and quantification of GFP/RFP-positive cells by flow cytometry after 3 days. TU = (P*N/100*V)*1/DF. ($p$ = % GFP + or RFP + cells, $N$ number of cells at the time of transfection, $V$ volume of dilution added to each well, $DF$ dilution factor).

**Inducible cleavage of RAD21-TEV**. For cleavage of RAD21-TEV, neurons were plated as described above and transduced at day 3 with lentivirus containing either ERt2-TEV or NLS-TEV at a multiplicity of infection of 1. For ERt2-TEV dependent RAD21-TEV degradation, neurons were treated on culture day 10 with 500 nM 4-hydroxytamoxifen (4-OHT) or vehicle (ethanol). For 24 h depletion and rescue of RAD21-TEV, NLS-TEV dependent RAD21-TEV degradation, neurons were treated on culture day 10 with 6-hour doxycycline (100 ng/ml) pulse or vehicle (water). Cells were then rinsed, and media replaced with conditioned neuronal media. For 7-day long-term NLS-TEV dependent RAD21-TEV degradation, neurons were treated on culture day 10 with 24-hour doxycycline (1 μg/ml) pulse or vehicle (water). Cells were then rinsed, and media replaced with conditioned neuronal media.

**Isolation of nuclei from post-mortem tissue**. Anonymized post-mortem human tissue was obtained from approved tissue banks and used in accordance with the Human Tissue Act (UK) and with approval by the Imperial College London Research Ethics Committee. Isolation of nuclei was performed as previously described[62] with minor modifications. Briefly, 50–250 mg of pre-frontal cortical grey matter was homogenized in 250 mM sucrose, 25 mM KCl, 5 mM MgCl2,

10 mM Tris pH 8.0, 1 μM DTT, 1X Proteinase Inhibitor w/o EDTA (Roche), 0.4 U μl-1 RNaseIn (ThermoFisher) 0.2 U μl-1 Superasin (ThermoFisher), 1 μM (DAPI) and centrifuged through an iodixanol gradient (Sigma). Pelleted nuclei were washed and then stained with NeuN antibody (Abcam, ab190195) in staining buffer (PBS, 1%BSA, 0.2 U/μl RNaseIn (Thermofisher)), NeuN antibody (1:200) for 1 h at 4 ºC. For negative controls antibody was excluded. Nuclei were then sorted on BD Fusion, and collected in wash buffer (PBS, 1%BSA, 0.2 U/μl RNaseIn (Thermofisher)) before RNA extraction.

**Protein analysis**. Whole cell extracts were prepared by suspending cells in PBS, centrifugation and resuspension in protein sample buffer (50 mM Tris-HCl pH6.8, 1% SDS, 10% glycerol) followed by quantification using Qubit (Invitrogen). Following quantification, 0.001% bromophenol blue, and 5% beta-mercaptoethanol were added. Sodium dodecyl sulphate-polyacrylamide gel electrophoresis (SDS-PAGE) was carried out with the Bio-Rad minigel system. Resolved gels were blotted on to polyvinylidene fluoride transfer membrane, followed by 1 h incubation in fluorescent blocker (Millipore), and then incubated in primary antibody diluted to the appropriate amount in fluorescent blocker (Millipore) overnight at 4 ºC. Primary antibodies were goat polyclonal to LAMIN B (1:5000, Santa Cruz Biotechnology, sc-6216), mouse monoclonal to LAMIN B (1:5000, Santa Cruz Biotechnology, sc-374015), rabbit polyclonal to NIPBL (1:1000; A301-779A, Bethyl Laboratories) mouse monoclonal anti-Myc tag for detection of RAD21-TEV (1:500, Santa Cruz biotechnology, sc-40), mouse monoclonal to NLGN1 (1:100, Santa Cruz biotechnology sc-365110) and goat polyclonal antibody to SYN1 (1:2500, Synaptic Systems, 106103). Blots were then incubated in secondary antibody diluted in fluorescent blocker (Millipore) for 1 h at room temperature. Secondary antibodies were donkey anti-goat IgG (H + L) Alexa Fluor 680 (1:10,000, Thermofisher, A-21804), goat anti-mouse IgG (H + L) Alexa Fluor 680 (1:10,000, Thermofisher, A-28183) and donkey-anti mouse IgG (H + L) Alexa Fluor 790 (1:10,000, Thermofisher, A-11371). Stained membranes were imaged on an Odyssey instrument (LICOR) and analysed using ImageStudioLite v5.2.5.

**Immunocytochemistry**. Neurons plated on coverslips were fixed with PBS containing 4% paraformaldehyde and 4% sucrose warmed to 37 ºC for 10 min at room temperature. Neurons were then permeabilized using 0.3% Triton X-100 for 10 min at room temperature, and blocked in blocking solution (10% normal goat serum, 0.1% Triton X-100 in PBS) for 1 h at room temperature. Samples were incubated with primary antibodies diluted in staining solution (0.1% Triton X-100, 2% normal goat serum in PBS) in a humidified chamber overnight at 4 ºC. Primary antibodies were mouse monoclonal TUJ1 (1:500, Biolegend, 801213), mouse monoclonal V5 (1:250, Sigma-Aldrich, V8012) and mouse monoclonal NeuN (1:1000, Abcam, ab104224). Samples were then washed with PBS and incubated in secondary antibodies diluted 1:500 in staining solution (0.1% Triton X-100, 2% normal goat serum in PBS) in a humidified chamber for 1 h at room temperature. Secondary antibodies were, goat anti-mouse IgG (H + L) Alexa Fluor 488 (ThermoFisher, A-1101) and goat anti-mouse IgG (H + L) Alexa Fluor 568 (Thermo-Fisher, A-11004). Cells were mounted in Vectashield medium containing DAPI (Vector Labs). Samples were visualized using a TCS SP5 Leica laser scanning confocal microscope using LAS X v2.7. Images were processed using Leica Confocal Software and FIJI. For quantification of NeuN-positive cells, DAPI-identified nuclei that colocalized with NeuN signal were counted as neuronal, and DAPI-identified nuclei without NeuN were counted as non-neuronal. Samples were quantified using a processing pipeline developed in CellProfiler (version 2.2, www.cellprofiler.org).

**Identification of human mutations**. NIPBL-mutation screening was performed by PCR of genomic DNA and Sanger Sequencing. Genomic DNA was isolated from brain cortical tissue using DNeasy blood & tissue extraction kit (Qiagen), according to the manufacturer's protocol. The entire NIPBL coding region (exons 2–47) was screened for mutations. Primer pairs were designed using ExonPrimer to amplify

exons, exon/intron boundaries, and short flanking intronic sequences. All PCR reactions were performed in a 25-μl reaction volume containing 60 ng of genomic DNA, 1 U of AmpliTaq Gold (Applied Biosystems), 20 pmol of each primer, 75 μM of each dNTP, 10X PCR buffer II (Applied Biosystems), and 1.0 mM or 1.5 mM of $MgCl_2$ (Applied Biosystems). Prior to sequencing all the PCR products were purified with USB ExoSAP-IT PCR Product Cleanup (Affymetrix) following the manufacturer's instructions, and subsequently sequenced using BigDye Terminator v3.1 cycle sequencing and analyzed on an ABI 3730 (Applied Biosystems, Carlsbad, CA). Sequence analysis was performed using Sequencher software version 5.2.4 and Mutation Taster[63].

**RNA extraction and sequencing**. RNA was extracted from both cells and nuclei using Picopure RNA Isolation Kit (Thermofisher) according to manufacturer's instructions. Residual DNA was eliminated using RNAse-Free DNase Set (Qiagen). Sorted nuclei were pelleted before proceeding with downstream RNA isolation. Cultured neurons were treated with extraction buffer (Picopure RNA Isolation Kit, Thermofisher) directly in the well before proceeding with downstream RNA isolation. RNA was assessed for quality (Bioanalyzer, Agilent) and quantity (Qubuit, Invitrogen). Libraries for RNA-sequencing of RAD21-TEV neurons were prepared using 200 ng of RNA and NEBNext Ultra II Directional RNA Library Prep Kit for Illumina with polyA enrichment for ERt2-TEV, and ribozero ribosomal RNA depletion by NLS-TEV. Libraries for RNA-sequencing of human cortical nuclei were prepared using 40 ng of RNA and NEBNext Ultra II Directional RNA Library Prep Kit for Illumina using ribozero ribosomal RNA depletion. Library quality and quantity were assessed on a bioanalyser and Qubit, respectively. Libraries were sequenced on an Illumina HiSeq2500 (v4 chemistry), generating 40-million paired-end 100 bp reads per sample.

**SLIC-CAGE**. SLIC-CAGE libraries were made using the published SLIC-CAGE protocol[64,65]. Briefly, 50 ng of nuclear RNA was used per sample and mixed with 4.95 μg of the RNA carrier mix. The library fragments derived from the carrier RNA were digested using I-CeuI and I-SceI homing endonucleases, and the library fragments derived from the sample RNA were PCR-amplified (14-15 amplification cycles). Eight samples with different barcodes were standardly multiplexed prior to sequencing and the resulting library mixes sequenced on a HiSeq2500 in single-end 50 bp mode.

**ATAC-seq**. ATAC-seq was performed on $5 \times 10^5$ nuclei isolated from control ($n = 5$) and CdLS patients ($n = 4$) per sample, following the omni-ATAC protocol[66]. Omni-ATAC is an improvement of the original ATAC protocol, with the inclusion of several additional detergents, NP-40, Tween-20 and digitonin in the nuclei digestion. After 30 min of incubation, samples were purified using the Zymo ChIP DNA Clean and Concentrator (Zymo Research) and libraries prepared as previously described[67].

**4C-seq**. 4 C was performed as previously described[39] with modifications. Briefly, neuronal cells were cross-linked in PBS with 1% formaldehyde for 10 min at RT and nuclei were isolated in lysis buffer (10 mM Tris-HCl pH 7.4, 150 mM NaCl, 0.5% NP-40, 1 x protease inhibitors). After digestion using HindIII, digested products were ligated by T4 DNA ligase. 3 C templates were purified and then digested again using a second enzyme, NlaIII. Final 4 C sequencing libraries were purified using a Roche High-Pure PCR Product Purification Kit, and then sequenced using an Illumina HiSeq 2500 platform. Sequence data was analysed using the 4 Cseqpipe software suite[68] with the setting of "-stat_type mean -trend_resolution 40,000". 4C-seq primers are listed in Supplementary Data 15.

**RNAseq analysis**. Paired-end 100 bp sequencing reads were aligned against mouse genome (mm9) or human genome (hg38) with Tophat2 (2.0.11)[69]. Reference sequence assembly and transcript annotation that was obtained from Illumina iGenomes. Gene based read counts were obtained using featureCounts function from Rsubread Bioconductor package (1.24.2)[70,71]. Normalization was performed in DESeq2 Bioconductor package (1.24.0)[72] and data was rlog transformed to allow for visualization by PCA and heatmaps. Differentially expressed gene (DEG) analysis was also performed with DESeq2 and DEGs were defined with Benjamini-Hochberg adjusted $P < 0.05$. GO term analysis was performed with goseq Bioconductor package (1.24)[73]. After converting mouse gene symbol to human gene symbol using Ensembl Biomart[74], Gene Set Enrichment Analysis (GSEA 2.2.0)[75,76] was performed with GseaPreranked tool using Hallmarks gene set. For RAD21-TEV RNAseq data, polyadenylated genes were annotated using PolyA_DB (ref. [77]) and non-polyadenylated genes were removed from the analysis. Glial genes were defined by control NeuN-negative versus control NeuN-positive RNAseq data with adj $P < 0.05$ and fold-change >20. We processed CdLS NeuN-positive RNAseq data after removing glial genes. In both human CdLS NeuN-positive and NeuN-negative RNAseq dataset, we only focused on autosomes to mitigate sex specific differences. Unwanted batch effects were controlled in CdLS NeuN-positive and NeuN-negative RNAseq by using R Bioconductor package RUVseq (ref. [78]) with RUVg function and $k = 2$ for both. For CdLS NeuN-negative RNAseq, sample CDL-744P was excluded. Heatmaps were generated using R Bioconductor package ComplexHeatmap (ref. [79]) and GraphPad Prism 8.0.

**SLIC-CAGE analysis**. Sequenced libraries were demultiplexed using CASAVA allowing zero mismatches for barcode identification. Demultiplexed CAGE tags (47 bp) were mapped to a reference GRCh37/hg19 human genome using Bowtie2 (--phred33-quals -U)[80] with default parameters allowing zero mismatches per 22 nucleotide seed sequence. The mapped reads were then sorted using Samtools v 1.10 (ref. [81]) and uniquely mapped reads kept for downstream analysis in R graphical and statistical computing environment (http://www.R-project.org/). The mapped and sorted unique reads were imported into R as bam files using the standard workflow within the CAGEr package v1.20 (ref. [82]). All 5′ends of reads are CAGE-supported transcription start sites (CTSSs) and the number of each CTSS (number of tags) reflects the transcript expression levels. Raw tags were normalized using a referent power-law distribution (alpha = 1.48) and expressed as normalized tags per million (TPMs)[83]. The highest expressed CTSS is termed the 'dominant CTSS'. The correlation of samples on CTSS level was estimated based on TPM values. This analysis led to the removal of sample CDL-744P due to its low correlation with the rest of the samples. CTSSs with sufficient support (at least 0.1 TPM in at least one sample) were further used to construct promoter regions. First, sample-specific clustering of CTSSs was performed to define tag clusters, allowing a maximum distance of 20 bp between any 2 CTSSs in the same tag cluster. In order to allow for between-sample expression profiling, tag clusters with sufficient support (at least 5 TPM) within 100 bp of each other were aggregated across all samples to define promoter regions (consensus clusters). To allow comparison with RNA-seq data, genomic coordinates of consensus clusters were converted into GRCh38/hg38 coordinates using the liftOver tool[84]. Consensus clusters were annotated using ChIPseeker v.1.22.1 (ref. [85]) with the transcript annotation *TxDb.Hsapiens.UCSC.hg38.knownGene* to define TSSs of known genes and the org.Hs.eg.db annotation to allocate Ensembl IDs to gene symbols. A consensus cluster was assigned to the promoter of a known gene if it was within a region 500 bp upstream and 100 bp downstream of the annotated TSS or within its 5′ UTR. This resulted in a set of 18,867 promoter-associated consensus clusters. To reduce the effect of sex differences, we excluded consensus clusters located on chromosomes X and Y. Consensus clusters were further filtered to exclude those associated with glial-specific genes, resulting in a final set of 18,103 consensus clusters and corresponding Ensembl IDs. DESeq2 v.1.26.0 (ref. [72]) was used to define differentially expressed genes, accounting for treatment and age effects. Genes with BH-adjusted $P < 0.05$ and $\log_2$ fold-change values of $> 0$ or $< 0$ were considered as significantly up- or down-regulated. Principal component analysis was implemented in DESeq2 on rlog-transformed read counts. PCA plots were created using the top 10,000 consensus clusters with the highest row variance. Variance associated with age group was removed from the rlog-transformed data with limma v.3.42.0 (ref. [86]) for visualization and plotted as described above. Functional enrichment analysis of significantly up- or downregulated genes was performed using GO over-representation test with clusterProfiler v.3.14.3 (ref. [87]). Enriched GO terms associated with biological processes were determined based on a BH-adjusted $P$ value cutoff = 0.05 and a $q$ value cutoff = 0.05 against a background of genes expressed in control samples (mean CAGE signal > 2 TPM, $n = 11,634$). CAGE data were visualized using the Integrative Genomics Viewer (Broad Institute)[88].

**ATAC-seq analysis**. After demultiplexing, raw reads were trimmed with Trim Galore! (trim_galore_v0.4.4; https://www.bioinformatics.babraham.ac.uk/projects/trim_galore) with arguments --trim-n --paired. The trimmed reads were then aligned to hg38 using bowtie2 (bowtie2/2.2.9, with -p8 -t --very-sensitive -X 2000)[80]. Samtools 1.2 (ref. [81]) was applied for sorting the reads and picardtools MarkDuplicates (1.90; Broad Institute 2019) was used for marking duplicated reads. R bioconductor package ATACseqQC (ref. [89]) was applied for removing duplicated reads, reads without properly mapped mates, and reads mapped to chrMT. alignmentSieve function from deeptools[90] was applied for shifting ATACseq reads[67] and keeping nucleosome-free reads (fragment length < = 100 bps) with argument --ATACshift --minFragmentLength 0 --maxFragmentLength 100. We used the nucleosome-free bin for the downstream analysis. Peak calling was proceeded with MACS2 peakcalling function with -f BAMPE -g hs (ref. [91]). Peaks overlapped with the black list hg38 v2 (ref. [92]) were removed and only those that appeared in more than any 2 samples were kept. Peak-based read counts were then obtained using the featureCounts (ref. [71]) function from Rsubread Bioconductor package. Differential accessibility analysis has proceeded with DESeq2 Bioconductor package[70]. Peaks were annotated with ChIPseeker Bioconductor package[85].

**Cohesin binding and enhancer proximity**. The binding of genes by cohesin and CTCF was defined by RAD21 (ref. [21]) (ENCODE ENCSR198ZYJ) or CTCF ChIP-seq (ENCODE ENCSR677HXC, ENCODE ENCSR822CEA) peaks overlapping with gene bodies. The proximity of genes to enhancers was defined by whether genes were the nearest neighbour to a neuronal enhancer[47,93]. R Bioconductor package GeneOverlap[94] was applied for one-tailed Fisher's exact test and reported the odds ratio and $P$ value.

**Reporting summary**. Further information on research design is available in the Nature Research Reporting Summary linked to this article.

## Data availability

The data that support this study are available from the corresponding author upon reasonable request. RNA-seq, ATAC-seq and 4C-seq data generated for this study have been deposited at GEO under accession number GSE150130. CAGE data generated for this study has been deposited at ArrayExpress under accession number E-MTAB-9045. Published datasets used in this study can be found at the following locations: Human Autism spectrum disorder gene expression, syn4587609, Human 15q duplication gene expression syn4587609, Mouse Neuron HiC GSE96107, RAD21 binding mouse neurons SRX3381923, CTCF binding mouse neurons ENCSR677HXC, Enhancers mouse neurons GSE60192, RAD21 binding human neurons ENCSR198ZYJ, CTCF binding human neurons ENCSR822CEA, Human neuron enhancers phs001373.v2.p2, Nipbl + /-mouse gene expression available as supplemental file from [https://journals.plos.org/plosgenetics/article?id=10.1371/journal.pgen.1000650#s4], Ezh1/2 KO mouse gene expression GSE84245, Mecp2 + /- mouse gene expression GSE113673, Fmr1-/- mouse gene expression GSEA 81912, Mouse Huntington's gene expression GSE65776, Pten mutant mouse gene expression GSE59318, Human Rett Syndrome gene expression GSE113673, Human Huntington's Disease gene expression available as supplemental file from [https://journals.plos.org/plosone/article?id=10.1371/journal.pone.0143563#sec021], Human Alzheimer's Disease gene expression syn18485175, Human Down Syndrome gene expression GSE59630. Source data are provided with this paper.

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

## Acknowledgements

We thank G. Little, M. Clements, and S. Parrinello, for advice and practical instruction, S. Di Giovanni and A. Schaefer (Icahn School of Medicine, NY) for discussion and comments on the manuscript, B. Patel, J. Elliott, and the LMS/NIHR Imperial Biomedical Research Centre Flow Cytometry Facility for cell sorting, D. Gomez-Cabrero for advice on batch correction of RNA-seq data, K. Tachibana (Institute of Molecular Biotechnology, Vienna) for providing the NLS-TEV vector, and A. Smith for advice on preparation and sorting of neuronal nuclei. Human brain tissues, and donor-associated de-identified demographic, clinical and phenotypic data used in this study were obtained from the NIH NeuroBioBank. All specimens were collected under the approval of Institutional Review Boards. A complete inventory is available at NIH NeuroBioBank [https://neurobiobank.nih.gov]. We thank and acknowledge the Edinburgh Brain and Tissue Bank for providing the tissues used in this study. The Edinburgh Brain Bank is supported by the MRC. Ethical approval was by the NHS-RECSE (16/ES/0084). This work was funded by the Medical Research Council UK, Wellcome (Investigator Award 099276/Z/12/Z to MM), EMBO and HFSP (HFSP LT00427/2013, EMBO ALTF 1047-2012 TO LC).

## Author contributions

F.D.W., L.C., Y.G., N.C. and M.K. did experiments, F.D.W., L.C., Y.G., Y.-F.W., R.G., Y.G., N.C., M.K. and G.D. analysed and curated data, F.D.W., L.C., R.G., M.K. and M.M. visualized data, F.D.W., L.C., I.D.K., B.L., A.G.F. and M.M. wrote the paper.

## Competing interests

The authors declare no competing interests.
