## [Peer Review File · Nature Communications]

REVIEWER COMMENTS

Reviewer #1 (Remarks to the Author):

In this work, Weiss et al. determine the transcriptome of cortical neurons from post-mortem tissue of Cornelia de Lange Syndrome (CdLS) patients (3 with NIPBL mutations and the 4th with unknown mutation), and found that a significant set of downregulated genes overlap with downregulated genes upon acute depletion of cohesin in mouse cells. They also show that restoring cohesin rescues expression of a significant number of genes. Thus, they conclude that cohesin is continuously required for neuronal gene expression and argue in favor of this approach for addressing mechanisms of neural dysfunction in CdLS.

Although it is an interesting work with certainly new data (the transcriptomes of cortical neurons of CdLS patients and of cohesin-depleted mouse neurons have not been previously addressed), I have several concerns.

- First, I am not sure about the message the authors want to transmit. Try they to demonstrate that CdLS derives from a deficit in cohesin? If so, several publications, not cited in this manuscript, support uncoupling of cohesin from NIPBL disruption effects, the major determinant of CdLS (Nolen et al (2013) Human Mol Genet 22: 4180; Remeseiro et al (2013) Biochim Biophys Acta 1832: 2097; Zuin et al (2014) Plos Genet 10: e1004153).

- Indeed, NIPBL mutations account for more than 60% of CdLS cases, while RAD21 mutations only for less than 1%. And these mutations are always in heterozygosis. In fact, NIPBL, when mutated, results in a slight reduction of protein levels (70-75% of the protein remains present). Why them authors decide to study the effects of deep cohesin depletion? Haven't they try to analyze transcriptome on the context of the RAD21-TEV allele in heterozygosis, or alternatively try to shorten cleavage time of RAD21 in order to have milder depletion conditions? To date, most data support that CdLS is not due to major cohesin deficiency, and point to roles of NIPBL localized at regulatory elements independent of cohesin. Whether this role is structural or not remains to be addressed. This is a crucial issue that authors must address.

- Authors find 307 genes downregulated in neurons from CdLS patients and 463 and 570 under cohesin depletion with ERT2 and NLS-TEV, respectively. Among these downregulated genes, 51 and 57 are common in patients and ERT2 and in patients and NLS-TEV, which is a significant overlap but far from full overlap. It is possible that whatever they alter, either cohesin or Nipbl, if this disturbs genome structure or transcription a similar impact in the transcriptome is expected. As authors indicate, expression of genes related to specialized neuronal functions use to depend on distal regulatory elements, and account for important transcriptional output of specialized cells. It is reasonable to expect that conducting cohesin depletion approaches, or depletion of other genome structural proteins non related to cohesion, in other organs during development, would probably lead to transcription alteration of genes related to specialized functions of these organs. Thus, similar transcriptional impact would be expected from different, non-related, ways of altering genomic structure, and this does not demonstrate that most CdLS cases, those linked to NIPBL mutations, are explained by cohesin deficiency. It would have been interesting to analyze neurons of Nipbl mutant mice, embryos and adults, and to study real cohesin deficiency at chromatin regions at both stages.

- At several points author wonder about whether effects in CdLS result of cohesin deficiency during development in forming neurons, or reflect an ongoing requirement for cohesin in mature neurons. This is an interesting point, since transcriptome of neural post-mortem tissue may not accurately reflect the major transcriptional defects at relevant developmental stages related to CdLS phenotypes. However, authors assume that it is always a problem of cohesin.
- Previous reports show that RAD21 depletion has no major consequences on general transcription, excepting some genes dependent on distal elements as indicated by authors. Some of these papers are not cited or discussed (Fisher et al (2017) *Leukemia* 31: 712, not cited; Schwarzer (2017) *Nature* 551: 51, cited in the introduction but not discussed). Similarly, some recent studies with neurodevelopment-related transcriptomic results in the context of CdLS are not cited or discussed (Luna-Pelaez et al (2019) *Cell Death Dis* 19:548, not cited; van den Berg (2017) *Neuron* 93: 348, cited in the introduction but not discussed).
- In view of my comments, title seems inaccurate, since authors are not exactly rescuing genes deregulated in CdLS by cohesin, they are rescuing genes deregulated under cohesin depletion conditions by cohesin restoration.
- Minor: on page 14, end of 1st paragraph, when authors refer to expression in embryonic Nipbl+/- brain, they indicate ref 14, and they should indicate ref 19.

Reviewer #2 (Remarks to the Author):

In this study by the Merkschlager lab, the authors analyze the transcriptome of cortical neurons from four CdLS patients and find that hundreds of genes involved in neuronal functions are downregulated. Importantly, they identify common transcriptional changes in CdLS and autism (a condition frequently observed in CdLS) but not with other neurological disorders (Huntington Disease or Alzheimer). Next, they compare the CdLS transcriptional changes with those found in cortical neurons from E14.5 murine embryos after depletion of cohesin Rad21 and find a remarkable overlap. Some results hint to altered gene organization in the cohesin-depleted cells as the cause of deregulated gene expression. Moreover, proper gene expression of deregulated genes can be observed after recovering normal Rad21 levels in these cells. From these results, the authors conclude that the presence of cohesin is continuously required for proper expression of genes important for the functionality of cortical neurons. This opens the door to postnatal interventions to restore, or at least improve, tissue/organ functionality in CdLS patients.

I think the study is very interesting and provides valuable data regarding the cause of neurological dysfunction in CdLS patients, as well as important support for the idea that the role of cohesin in gene regulation is at the origin of CdLS pathology. The paper is written in a succinct format and it is sometimes difficult to understand how the analyses were performed. Maybe the Methods section should be extended and the datasets used for comparisons of gene deregulation in other mouse models and other neurological disorders should be indicated more explicitly.

One concern is that the authors compare human adult brain neurons with mouse embryonic brain neurons, the former mutant for NIPBL in heterozygosity (at least 3 patients, the fourth has an unidentified mutation) and the latter almost completely depleted of cohesin. Three questions here:

- 1- How similar are the transcriptomes of these neurons (adult human versus mouse embryo)?
- 2- How do the NIPBL mutations identified in patients affect cohesin's presence on chromatin in general and at particular genome locations? Ideally, this should be assessed in patients' samples, but alternatively the authors could try some other cell model.
- 3- Authors identify around 50 genes deregulated in both CdLS and Rad21-depleted neurons and further show that re-expression of cohesin in the murine neurons restores expression of many of those genes. What is the relevance of these genes misexpression to neuron malfunction in CdLS? For instance, are these genes also deregulated in ASD?

Additional issues

4- Patient selection: do the patients used in this study present similar neurological symptoms (ASD, intellectual disability?). Are the patients harboring NIPBL mutations more related (transcriptionally/clinically with ADS patients when compared with the patient harboring an unidentified/cohesin unrelated mutation? Please comment. PCA for CAGE and RNA seq analyses of patient and control samples should be shown.

5- "NIPBL RNA levels were not significantly deregulated by the NIPBL loss of function mutations in these patients (Extended Data Fig. 1d)". The authors should try allele-specific RT-qPCR in patient samples to check expression of wt and mutant alleles. Ideally, it would be important to check also protein levels in the patient cells (cohesin and NIPBL).

6- "Genes that were downregulated in response to acute cohesin depletion in primary neurons were significantly enriched for the binding of RAD21 and CTCF ... and for proximity to enhancers" (Figure 2i) What are the source data for these analyses? In Figure 3c a similar analysis is performed for CdLS deregulated genes. In Methods the authors indicate they use data from human neural cells. Are human cell data also used in the analyses in 2i? Are there no data in mouse?

7- In addition to the comparisons with other mouse models of neurological disorders, the authors should compare the transcriptome of the Rad21 depleted neurons to the data of Smc3^{+/-} adult mice data from Fujita 2017. It may be also interesting to compare to the STAG1 KO E17.5 embryos from Cuadrado 2015.

8- Upregulated versus downregulated genes. The authors highlight the idea that cohesin is important for proper transcription of genes that depend on distal enhancers (as opposed to genes that are "promoter centric"). It is unclear whether this is true both for activation and repression of transcription. In CdLS there are as many genes upregulated as downregulated, but overall the upregulated genes do not show significant GO term enrichment, show lower odd ratio for cohesin, CTCF occupancy and enhancer proximity, etc. The authors should discuss their views on the mechanism of upregulation.

9 -In Discussion

“In addition to directly cohesin-dependent genes, reduced cohesin function in CdLS neurons also affected genes that were not sensitive to acute cohesin depletion in mouse neurons. These lacked hallmarks of direct cohesin targets.”

Cryptic. The concept of cohesin-target gene needs to be clarified. The fact is that most genes affected in CdLS are not affected in mouse neurons after Rad21 depletion. Also, there is no formal proof of “reduced cohesin function” in CdLS neurons in the current manuscript (see question 3 above).

- The authors could consider (and even discuss) the possibility that some deregulated genes in CdLS patients are the consequence of a role of NIPBL independent of cohesin. Moreover, it is formally possible that (most, several, some) genes found deregulated in both CdLS patients carrying NIPBL mutations and Rad21-depleted mouse neurons are regulated by NIPBL and cohesin through independent mechanisms.

Reviewer #3 (Remarks to the Author):

In this manuscript, Weiss et al. perform a series of transcriptional profiling experiments and associated data analysis to shed light into the contribution of the cohesin complex to the molecular pathogenesis of Cornelia de Lange Syndrome. First, they profile the transcriptome in neuronal nuclei collected from human Cornelia de Lange Syndrome patients, demonstrating a signature of gene expression misregulation consistent with other neurodevelopmental disorders. Next, the authors engineer a system to acutely degrade RAD21, a subunit of cohesin, via TEV protease cleavage in primary mouse neurons. Upon rapid loss of RAD21/cohesin, using different methods of inducible TEV protease expression, the authors observe a swathe of gene expression changes, a significant subset of which overlap with the patient-derived data. These data demonstrate that sustained function of the cohesin complex is required for normal neuronal gene expression. Last, the authors allow for reformation of cohesin and show that most of the gene expression changes are corrected. Overall, this study is interesting and demonstrates the utility of exploring the consequences of disease-causing genes/complexes outside the context of germline mutation or deletion; however, some facets of this work require further investigation to benchmark the tools and provide further insight into the molecular descriptions explored in this manuscript.

Major points:

1. Given the widespread disturbance of neuronal genes upon acute RAD21 cleavage, it would be important to understand if sustained RAD21 cleavage affected neuronal maturation and function. If sustained RAD21 depletion does not result in downstream neuronal dysfunction, then the functional impact of these acute gene expression changes is unclear, as we know the mouse model demonstrates these deficiencies. For example, the authors could measure neuronal morphology, synaptic marker

changes at the protein level, or electrophysiology to show the gene expression changes have an impact on the neuronal biology. These experiments will be important in benchmarking this tool as a method to study cohesin, neuronal function, and pathogenesis of CdLS.

2. Along these lines, does sustained RAD21 depletion (for example for 10-14 days) cause further gene expression changes, and to what extent do these gene expression changes quantitatively align with the human data relative to the 24-hr depletion signature? The authors should measure the transcriptome in long-term RAD21 depleted cultures.

3. Does the genomic architecture return to the wild-type state upon rescue of cohesin? Or does the architecture of the genome adapt to a new steady state that is somewhat different than an unperturbed neuron. If so, do these differences account for some of the gene expression changes that do not rescue or that occur de novo upon RAD21-depletion? The authors can perform chromosome conformation capture (ideally Hi-C) in the RAD21-rescue experiment to address this question.

4. Cohesin is an important factor for all cell types. How the cohesin-dependent signature changes amongst cell-types would help our understanding of the importance of genomic architecture for neural cells. For example, the authors can directly compare their NeuN+ and NeuN- gene expression data in humans to see what is similar/different between the cell types. (particularly as the authors observe the NeuN- population exhibits far more gene expression changes than the neuronal population).

Additionally, the authors should also compare their transcriptomic studies to the work in Rao et al. (PMID: 28985562), who also perform acute depletion of cohesin in a human cell line, and also observed wide spread transcriptome misregulation. Exploring of these intersections will help hone in on the specific importance of cohesion in the brain as the overlap between the CdLS human data and mouse acute cohesin depletion was minimal, albeit statistically significant.

Minor points:

1. It would be informative if the authors would include a reduced dimension presentation (e.g., PCA) of the human patient gene expression data. This would help demonstrate how similar the gene expression signature is of the patient with CdLS but no identified CdLS mutation and the others.

2. Could the authors provide a brief description in each data table? Because there are so many, it would help future readers quickly orient themselves to the data and comparisons at hand. Particularly if they were to download the whole supplement and not individual tables.

Response to referees

We thank the referees for their thoughtful and constructive comments. We were pleased that the referees found the study interesting, and were appreciative of the novelty of the data. We acknowledge that the referees raised important points that needed to be addressed both experimentally and by textual revisions. Here we provide a brief overview of the new data and other changes we have made to the manuscript during revision, followed by a point-by-point response to each referee.

The revised manuscript contains new data and analysis in several key areas

1) Deeper characterisation of the RAD-TEV depletion system

a) We have extended the analysis of neurons 24h after RAD21-TEV depletion by new RNA-seq data on continued cohesin depletion for 3 and 7 days. These data show that neurons do not simply adapt to the loss of cohesin. In fact, gene expression changes become more extensive with extended time after the depletion cohesin. This is critical for the interpretation of the rescue experiments, which form a key part of the paper. These new experiments allow us to conclude that the reversal of gene expression we report indeed require the restoration of cohesin.

b) We also show that changes in gene expression translate into changes in protein expression. In this context we discuss data from a parallel study that demonstrates neuronal maturation defects in response to the long-term genetic depletion of cohesin in immature post-mitotic neurons.

c) Finally, to better understand the role of cohesin versus NIPBL in neuronal gene expression, we have performed additional RNA-seq experiments in Nipbl^{+/-} mouse neurons. This enables us to directly compare gene expression in 24h RAD21-depleted and Nipbl^{+/-} mouse neurons. We find extensive overlap in deregulated genes between 24h RAD21-depleted and Nipbl^{+/-} mouse neurons ($P < 2.2e-16$, Odds ratio = 7.56), indicating that neurons show similar transcriptional changes in response to cohesin depletion and Nipbl heterozygosity.

Overall, these new data and analyses greatly strengthen the manuscript and its conclusions.

2) Textual revisions

Based on the new data and the referees' comments we have made extensive revisions to the narrative. In particular, we have revised the introduction to provide a more balanced account of current knowledge about the relationship between NIPBL and cohesin. The results section has been modified to include the new data and analyses described above, and the discussion puts these results into context, avoiding any implicit suggestion that experimental cohesin depletion mimics all aspects of gene deregulation in CdLS. Overall, these revisions have clarified the four key points made by our paper:

- 1) Hundreds of genes enriched for neuronal functions related to synaptic transmission, signalling processes, learning and behaviour are deregulated in cortical neurons from CdLS patients
- 2) Cohesin is continuously required for neuronal gene expression
- 3) The genes affected by acute depletion of cohesin belonged to highly similar gene ontology classes and showed significant numerical overlap with genes deregulated in CdLS.
- 4) Reconstitution of cohesin function after acute cohesin depletion largely rescues altered gene expression, including the expression of genes that are also deregulated in CdLS.

Changes are marked in red in the revised manuscript.

Below, referee comments are in black, our response is in grey, and red sections are excerpts from the revised manuscript.

Referee 1.

We thank the referee for providing insightful and constructive comments and are pleased that referee judged the work as interesting and appreciated the novelty of providing transcriptomes of cortical neurons of CdLS patients and of cohesin-depleted mouse neurons.

First, I am not sure about the message the authors want to transmit. Try they to demonstrate that CdLS derives from a deficit in cohesin?

We thank the referee for asking us to more succinctly define the questions we pose in our manuscript. We have made extensive textual changes and performed additional experiments and analysis, as outlined below.

The introduction now provides a more balanced account of current knowledge about the relationship between NIPBL and cohesin, taking care to cite key studies listed by the referee: "Heterozygous or hypomorphic germline mutations of cohesin and associated factors such as the cohesin loading factor NIPBL result in reduced cohesin function¹⁴⁻¹⁷. Although compatible with normal cell cycle progression, such mutations result in a human developmental disorder known as Cornelia de Lange Syndrome (CdLS)¹⁸⁻²⁰. All CdLS patients show a degree of intellectual disability, and 60 to 65% have autism spectrum disorder (ASD), often in the absence of structural brain abnormalities or neurodegeneration^{19,21,22}.

Experimental perturbations of neuronal cohesin and cohesin-related factors such as Nipbl during mouse development have shown changes in animal behaviour as well as abnormal neuronal morphology²³, migration²⁴ and gene expression²³⁻²⁶. NIPBL has historically been considered as separate from the core cohesin complex, and with distinct^{24,28} and sometimes even antagonistic activities²⁹. However, current structural³⁰ and functional³¹ evidence indicates that NIPBL is integral to the cohesin complex in its loading state³⁰ and contributes to ATPase activity³¹ during loop extrusion, the process thought to form chromatin loops, TADs and contact domains⁵. Accordingly, the depletion of NIPBL and cohesin have similar or identical effects on 3D genome organisation^{7,8}. Nevertheless, it remains unclear to what extent gene expression changes in cells with *Nipbl* mutations relate to cohesin versus potential other functions of NIPBL^{24,28,29}."

The revised Results section presents new experimental data (RNA-seq of *Nipbl*^{+/-} mouse neurons) and analysis (analysis of new RNA-seq of *Nipbl*^{+/-} mouse neurons and of published microarray data for whole *Nipbl*^{+/-} mouse brain): "There was significant overlap between changes in gene expression in human CdLS with published²⁵ gene expression in *Nipbl* heterozygous mouse brain at embryonic day 13.5 (e13.5; *Nipbl*^{+/-} whole brain versus human CdLS NeuN positive and NeuN negative nuclei combined; $P = 1.32e-14$, Odds ratio = 2.05) and with gene expression in *Nipbl*^{+/-} cortical neurons in explant culture (*Nipbl*^{+/-} cortical neurons versus human CdLS NeuN positive neurons; Supplementary Data 7; $P = 8.38e-06$, Odds ratio = 3.03). "

The RNA-seq experiments in *Nipbl*^{+/-} mouse neurons enabled us to directly compare gene expression in 24h RAD21-depleted and *Nipbl*^{+/-} mouse neurons. We found extensive overlap in deregulated genes between 24h RAD21-depleted and *Nipbl*^{+/-} mouse neurons ($P < 2.2e-16$, Odds ratio = 7.56), indicating that neurons show similar transcriptional changes in response to cohesin depletion and *Nipbl* heterozygosity: "There was significant overlap between genes deregulated by acute cohesin depletion with genes deregulated in embryonic *Nipbl*^{+/-} brain²⁵ and *Nipbl*^{+/-} RNA-seq in neurons, but not with genes deregulated in other mouse models of neuronal dysfunction⁴⁰⁻⁴⁴." The data are shown in the revised Supplementary Fig. 7.

Supplementary Figure 7. Neuronal gene deregulation in acute RAD21-TEV depletion, *Nipbl* heterozygosity and other mouse models of neuronal dysfunction

a) Representative fluorescent western blot for NIPBL and LAMINB proteins (left). NIPBL protein levels quantified by fluorescent western blot analysis and represented as relative to *Nipbl*^{+/+}. LAMINB protein levels were used for normalization (right, mean and SD of $n = 7$ biological replicates).

b) Volcano plot of gene expression fold-change versus adjusted P value of up- and downregulated genes in *Nipbl*^{+/-} neuronal explant cultures analysed by RNA-seq after 10 days in vitro ($n = 3$ biological replicates). At adj $P < 0.05$, 69 genes were up- and 251 genes were downregulated in *Nipbl*^{+/-} versus *Nipbl*^{+/+} neurons.

c) Gene ontology of biological functions for downregulated and upregulated genes in *Nipbl*^{+/-} neurons

d) Bar graph of overlap between deregulated genes in RAD21-TEV neurons and animal models of neuronal dysfunction including *Nipbl*^{+/-} neurons (this study) *Nipbl*^{+/-} brain²⁵, deletion of PRC2 components *Ezh1* and *Ezh2* (*Ezh1-2*^{-/-})⁴⁰, *Mecp2*^{+/-} (ref. 41), *Fmr1*^{-/-} (ref. 42), mutant *Htt* (*mHtt*), a mouse model of Huntington's disease⁴³ and mutant *Pten*^{m3m4/m3m4} (*Pten*^{mut}, ref. 44). P values were determined by Fisher's exact test.

e) GSEA showing the expression of in RAD21-TEV neurons of genes downregulated *Nipbl*^{+/-} neurons (DEseq2, adj. $P < 0.05$) and in embryonic *Nipbl*^{+/-} brain²⁵ (DEseq2, adj. $P < 0.05$).

The discussion puts these results into context and we now avoid any claim to the effect that experimental cohesin depletion mimics all aspects of gene deregulation in CdLS: "To explore the role of cohesin in neuronal gene expression we established in vitro models for the acute depletion of cohesin in primary cortical mouse neurons. Proteolytic cleavage of the essential cohesin subunit RAD21 disrupted 3D organization and perturbed the expression of neuronal genes. These experiments establish that cohesin is continuously required to sustain neuronal gene expression.

We found significant overlap between cohesin-dependent genes in mouse neurons and the neuronal transcriptome in human CdLS. Cohesin-dependent genes were enriched for cohesin binding and enhancer proximity in human neurons."

The revised abstract provides a succinct description of the findings: "We characterized the transcriptional profile of cortical neurons from CdLS patients and found deregulation of hundreds of genes enriched for neuronal functions related to synaptic transmission, signalling processes, learning and behaviour. Inducible proteolytic cleavage of cohesin disrupted 3-D genome organization and transcriptional control in post-mitotic cortical mouse neurons, demonstrating that cohesin is continuously required for neuronal gene expression. The genes affected by acute depletion of cohesin belonged to similar gene ontology classes and showed significant numerical overlap with genes deregulated in CdLS. Interestingly, reconstitution of cohesin function largely rescued altered gene expression, including the expression of genes deregulated in CdLS."

The message of the paper therefore is four-fold:

- 1) Hundreds of genes enriched for neuronal functions related to synaptic transmission, signalling processes, learning and behaviour are deregulated in cortical neurons from CdLS patients
- 2) Cohesin is continuously required for neuronal gene expression
- 3) The genes affected by acute depletion of cohesin belonged to highly similar gene ontology classes and showed significant numerical overlap with genes deregulated in CdLS.
- 4) Reconstitution of cohesin function after acute cohesin depletion largely rescues altered gene expression, including the expression of genes that are also deregulated in CdLS.

Why do the authors decide to study the effects of deep cohesin depletion? Haven't they tried to analyze transcriptome in the context of the RAD21-TEV allele in heterozygosis, or alternatively try to shorten cleavage time of RAD21 in order to have milder depletion conditions? To date, most data support that CdLS is not due to major cohesin deficiency, and point to roles of NIPBL localized at regulatory elements independent of cohesin. Whether this role is structural or not remains to be addressed. This is a crucial issue that authors must address

Our rationale for deep depletion of cohesin was to obtain a clear and definitive answer to the question whether cohesin is continuously required for neuronal gene expression. We subsequently determined that deep (85%, ERT2-TEV) and moderate (70%, NLS-TEV) cohesin depletion have qualitatively very similar impact on neuronal gene expression ($P < 2.2 \times 10^{-16}$, Odds ratio = 20.35). Importantly, the impact of acute cohesin depletion on gene expression overlaps significantly with the impact of NIPBL mutations both in human and mouse neurons ($P = 8.37 \times 10^{-10}$, Odds ratio = 2.74 and $P < 2.2 \times 10^{-16}$, Odds ratio = 7.56, respectively).

The open question whether or not NIPBL has additional functions at gene regulatory elements is not answered by our study, but this does not detract from the fact that loss of cohesin and mutations in NIPBL deregulate similar genes, and that the expression of these genes can be largely restored by reconstituting cohesin.

Authors find 307 genes downregulated in neurons from CdLS patients and 463 and 570 under cohesin depletion with ERT2 and NLS-TEV, respectively. Among these downregulated genes, 51 and 57 are common in patients and ERT2 and in patients and NLS-TEV, which is a significant overlap but far from full overlap ... Thus, similar transcriptional impact would be expected from different, non-related, ways of altering genomic structure, and this does not demonstrate that most CdLS cases, those linked to NIPBL mutations, are explained by cohesin deficiency

We agree with the referee that the observed overlap in gene expression between RAD21 depletion and CdLS is partial. However this overlap was significantly greater for CdLS than other afflictions of the human CNS, indicating that different perturbations have distinct effects on neuronal gene expression. Moreover, we have generated RNA-seq data in mouse neurons, and the deregulated genes again extensively overlap with those deregulated by

RAD21 depletion, much more so than other perturbations in mouse neurons (revised Supplementary Fig. 7a, please see above). We agree with the referee's comment that the biological effects of NIPBL mutations may not be fully explained by cohesin deficiency.

It would have been interesting to analyze neurons of *Nipbl* mutant mice, embryos and adults, and to study real cohesin deficiency at chromatin regions at both stages.

We thank the referee for this suggestion and have added RNA-seq data for *Nipbl*^{+/-} neurons to the paper (revised Supplementary Fig. 7a, please see above).

Referee 2.

We thank the referee for providing insightful and constructive comments. We were pleased to hear that 'the study is very interesting and provides valuable data regarding the cause of neurological dysfunction in CdLS patients, as well as important support for the idea that the role of cohesin in gene regulation is at the origin of CdLS pathology'

Maybe the Methods section should be extended and the datasets used for comparisons of gene deregulation in other mouse models and other neurological disorders should be indicated more explicitly.

We thank the referee for this valuable suggestion. We have added the requested table of all newly generated and previously published data sets analyzed in this study (Data Summary, Related Manuscript File).

One concern is that the authors compare human adult brain neurons with mouse embryonic brain neurons, the former mutant for NIPBL in heterozygosity (at least 3 patients, the fourth has an unidentified mutation) and the latter almost completely depleted of cohesin. Three questions here:

1- How similar are the transcriptomes of these neurons (adult human versus mouse embryo)?

Thank you for raising the important question. Gene expression is remarkably similar in mouse embryonic and adult human neurons: Adult human NeuN+ neurons have 15966 active genes, 12079 of which have a mouse homolog. Of these, 90.9% (10981) are also expressed in mouse RAD21-TEV neurons. Embryonic mouse RAD21-TEV neurons have 15739 active genes, 12948 of which have a human homolog. Of these, 85% (11024) are also expressed in NeuN+ neurons from adult human brain. Finally, ERT2-TEV 24h DE genes are enriched among genes that are expressed in both human and mouse (Odds ratio = 1.5, $P = 0.00042$).

To more deeply address the referee's question we have examined the overlap in deregulated gene expression between heterozygous *Nipbl* deficiency in embryonic mouse brain (PLoS 2009) and adult human CdLS neurons. We have also carried out additional RNA-seq experiments on *Nipbl*^{+/-} cortical neurons. Both comparisons showed significant

overlap. The revised manuscript reports these findings as follows: " There was significant overlap between changes in gene expression in human CdLS with published ²⁵ gene expression in *Nipbl* heterozygous mouse brain at embryonic day 13.5 (e13.5; *Nipbl*^{+/-} whole brain versus human CdLS NeuN positive and NeuN negative nuclei combined; $P = 1.32e-14$, Odds ratio = 2.05) and with gene expression in *Nipbl*^{+/-} cortical neurons in explant culture (*Nipbl*^{+/-} cortical neurons versus human CdLS NeuN positive neurons; Supplementary Data 7; $P = 8.38e-06$, Odds ratio = 3.03)". These data are shown in the revised Supplementary Fig. 7.

To further compare gene expression changes in response to cohesin cleavage versus heterozygosity in *Nipbl* we performed gene set enrichment analysis. There was overwhelming coherence between genes downregulated *Nipbl*^{+/-} neurons and in embryonic *Nipbl*^{+/-} brain ²⁵. These data are shown in the revised Supplementary Fig. 7e.

Supplementary Figure 7. Neuronal gene deregulation in acute RAD21-TEV depletion, *Nipbl* heterozygosity and other mouse models of neuronal dysfunction

a) Representative fluorescent western blot for NIPBL and LAMINIB proteins (left). NIPBL protein levels quantified by fluorescent western blot analysis and represented as relative to *Nipbl*^{+/-}. LAMINIB protein levels were used for normalization (right, mean and SD of n = 7 biological replicates).

b) Volcano plot of gene expression fold-change versus adjusted P value of up- and downregulated genes in *Nipbl*^{+/-} neuronal explant cultures analysed by RNA-seq after 10 days in vitro (n = 3 biological replicates). At adj *P* < 0.05, 69 genes were up- and 251 genes were downregulated in *Nipbl*^{+/-} versus *Nipbl*^{+/+} neurons.

c) Gene ontology of biological functions for downregulated and upregulated genes in *Nipbl*^{+/-} neurons

d) Bar graph of overlap between deregulated genes in RAD21-TEV neurons and animal models of neuronal dysfunction including *Nipbl*^{+/-} neurons (this study) *Nipbl*^{+/-} brain²⁵, deletion of PRC2 components *Ezh1* and *Ezh2* (*Ezh1-2*^{-/-})⁴⁰, *Mecp2*^{+/-} (ref. 41), *Fmr1*^{-/-} (ref. 42), mutant *Htt* (*mHtt*), a mouse model of Huntington's disease⁴³ and mutant *Pten*^{m3m4/m3m4} (*Pten*^{mut}, ref. 44). *P* values were determined by Fisher's exact test.

e) GSEA showing the expression of in RAD21-TEV neurons of genes downregulated *Nipbl*^{+/-} neurons (DEseq2, adj. *P* < 0.05) and in embryonic *Nipbl*^{+/-} brain²⁵ (DEseq2, adj. *P* < 0.05).

Finally, we have performed a three-way comparison of differential gene expression in human CdLS, mouse *Nipbl*^{+/-} brain and mouse RAD21-TEV cortical neurons, which is presented here:

Overlap of differential gene expression in RAD21-TEV, embryonic *Nipbl*^{+/-} mouse brain and adult human CdLS.

2- How do the NIPBL mutations identified in patients affect cohesin's presence on chromatin in general and at particular genome locations? Ideally, this should be assessed in patients' samples, but alternatively the authors could try some other cell model.

We thank the referee for raising this point, which we had neglected in the original version of the manuscript. NIPBL mutations are well documented to impact on the association of cohesin with chromatin, both in CdLS patient cells (Liu et al., 2009, revised ref. 14) and in model systems (revised references 14-17). The revised introduction now references this

substantial body of work. For example, Remeseiro et al. 2013 have shown that reduced NIPBL function leads to reduced SMC1 binding at gene promoters, especially promoters with reduced expression in *Nipbl*^{+/-} cells. Similarly, Chien et al., 2011 and Newkirk et al. 2017 have documented reduced local and genome-wide cohesin binding in *Nipbl*^{+/-} cells.

3- Authors identify around 50 genes deregulated in both CdLS and Rad21-depleted neurons and further show that re-expression of cohesin in the murine neurons restores expression of many of those genes. What is the relevance of these genes misexpression to neuron malfunction in CdLS? For instance, are these genes also deregulated in ASD?

We have performed the requested analysis and found that of the 81 genes that were deregulated CdLS and in at least one of the two RAD21-TEV depletion systems, 23 are deregulated in ASD or considered ASD risk genes. Their functions relate to cell adhesion, signalling, ion channels, synaptic transmission, transcription, and sphingolipid metabolism. We have added a Supplementary Table depicting these genes, their regulation in CdLS and experimental systems, function, and rescue to the revised manuscript (Supplementary Table 1). Restoration of cohesin levels rescued all 57 genes that were deregulated both in CdLS and in response to NLS-TEV-mediated cohesin depletion. We have added this information to the revised manuscript: " Of the 81 genes that overlapped between CdLS and one or both of the RAD21-TEV depletion systems, 23 were ASD risk genes³⁹ or deregulated in ASD (Ref. 35). These shared ASD genes mediate important neuronal functions in cell adhesion (*CLMP, FAT1, PCDHB10*) signalling (*PRKD1, NXPH3, FZD1, PHLDA1, PMEPA1, CAMK2G*), ion channels (*ATP1B1, KCNA1*), synaptic transmission (*LGI2, GABRA1, GABRG3, NETO2*), transcription (*SOX5, CECR2, CELF6*) and sphingolipid metabolism (*ST8SIA2*) (Fig. 3b, Supplementary Table 1)" and " Interestingly, all 57 genes that were deregulated both in CdLS and in response to NLS-TEV-mediated cohesin depletion were rescued by restoring cohesin expression. These included 14 genes implicated in ASD and shared between CdLS and NLS-TEV (Supplementary Table 1). Rescue was near-complete, regardless of the direction and the degree of the initial deregulation (Fig. 4d)".

4- Patient selection: do the patients used in this study present similar neurological symptoms (ASD, intellectual disability?). Are the patients harboring NIPBL mutations more related (transcriptionally/clinically with ADS patients when compared with the patient harboring an unidentified/cohesin unrelated mutation? Please comment. PCA for CAGE and RNA seq analyses of patient and control samples should be shown.

We thank the referee for this suggestion. Clinical records relating to the brain samples examined here indicate that all four patients had cognitive impairment. In addition, three of the patients had confirmed ASD features (CDL380P, CDL744P, CDL764P). We have no information on ASD status for 2082. We have added this information to the revised manuscript. We have included the requested PCA analyses for both CAGE and RNA-seq data as Supplementary Figures (Supplementary Fig. 2b, Supplementary Fig. 3a and Supplementary Fig. 4a). The data show that CdLS are separated from control samples by the major principal component, PC1. PC1 does not separate the three samples with identified NIPBL mutations from the single sample without an identified mutation.

Supplementary Figure 2. CAGE reveals deregulated gene expression in CdLS neurons

b) Principal component analysis of the 3 CdLS and 6 control samples successfully analysed by CAGE-seq (left) and volcano plot (right) of \log_2 fold-change versus adjusted P -value obtained from CAGE analysis of 3 patient and 6 control NeuN-positive samples. 408 down- and 358 upregulated genes were identified (adj. P -value < 0.05). Red indicates differential expression (DE).

Supplementary Figure 3. Analysis of NeuN-positive RNA-seq data related to Fig. 1.

a) Principal component analysis of the 4 CdLS and 6 control samples of NeuN-positive nuclei successfully analysed by RNA-seq.

Supplementary Figure 4. RNA-seq of NeuN-negative nuclei from CdLS cortex

a) Principal component analysis of the 3 CdLS and 6 control samples of NeuN-negative nuclei successfully analysed by RNA-seq (left) and volcano plot of gene expression fold-change versus adjusted P -value of up- and downregulated genes (right). 1397 genes were down- and 958 genes were upregulated (RUVg $k=2$, adj. $P < 0.05$, shown in red, DE = differentially expressed)

5- “NIPBL RNA levels were not significantly deregulated by the NIPBL loss of function mutations in these patients (Extended Data Fig. 1d)”. The authors should try allele-specific RT-qPCR in patient samples to check expression of wt and mutant alleles. Ideally, it would be important to check also protein levels in the patient cells (cohesin and NIPBL).

NIPBL mRNA expression changes are usually subtle, and not always visible in cells with heterozygous *NIPBL* mutations (Liu et al., 2009). Based on a sample of 4 primary tissues, failure to detect significant differences in mRNA expression is not unexpected. Moreover, analysis of total and allele-specific mRNA previously documented the lack of nonsense-mediated decay despite reduced protein expression in patient-derived lymphoblastoid cell lines, even though the mutations lead to reduced NIPBL protein expression (Parenti et al., 2020, <https://doi.org/10.1016/j.celrep.2020.107647>). Nevertheless, as detailed in response to point 2, NIPBL mutations reduce the association of cohesin with chromatin, both in CdLS patient cells (Liu et al., 2009, revised ref. 14) and in model systems (revised references 14-17). We have therefore decided to remove the figure panel showing mRNA expression. However, we now show NIPBL western data in the context of new RNA-seq data for *Nipbl*^{+/-} neurons as part of the revised Supplementary Fig. 7 (please see above)

6- “Genes that were downregulated in response to acute cohesin depletion in primary neurons were significantly enriched for the binding of RAD21 and CTCF ... and for proximity to enhancers” (Figure 2i)

What are the source data for these analyses?

In response to the referee's comment we have added a new table that shows the data generated in this study as well as external data used for comparison (Data Summary, Related Manuscript File).

In Figure 3c a similar analysis is performed for CdLS deregulated genes. In Methods the authors indicate they use data from human neural cells. Are human cell data also used in the analyses in 2i? Are there no data in mouse?

We thank the referee for spotting this error. We have corrected the legend to 'mouse enhancers'. The source of the data is again shown in the Data Summary (Related Manuscript File).

7- In addition to the comparisons with other mouse models of neurological disorders, the authors should compare the transcriptome of the Rad21 depleted neurons to the data of Smc3^{+/-} adult mice data from Fujita 2017.

Fujita et al., 2017 (Ref. 17) saw deregulated gene expression only at a single time point (P7). As shown in the table below, the gene expression data from Fujita et al., 2017 (Ref. 17) represents an outlier compared to *Nipbl* brain (Ref 25), *Nipbl* neurons (this study), *Stag1* (Ref 15), and CdLS (this study).

		ERT2-TEV		NLS-TEV	
		Odds Ratio	P-value	Odds Ratio	P-value
CdLS NeuN+	This study	2.43	1.49E-07	2.74	8.37E-10
Nipbl ^{+/-} neurons	This study	7.56	1.18E-31	6.8	8.87E-26
Nipbl ^{+/-} brain	Ref 25	4.02	1.18E-30	6.08	1.11E-56
STAG1 ^{-/-}	Ref 15	3.31	6.11E-24	5.58	8.03E-56
SMC3 ^{+/-}	Ref 23	0.3	0.997	1.14	0.376

Comparison of gene expression data from experimental models for cohesin- and NIPBL-depletion

8- Upregulated versus downregulated genes. The authors highlight the idea that cohesin is important for proper transcription of genes that depend on distal enhancers (as opposed to

genes that are “promoter centric”). It is unclear whether this is true both for activation and repression of transcription. In CdLS there are as many genes upregulated as downregulated, but overall the upregulated genes do not show significant GO term enrichment, show lower odd ratio for cohesin, CTCF occupancy and enhancer proximity, etc. The authors should discuss their views on the mechanism of upregulation.

We agree with the referee's assessment that the upregulated genes are more heterogeneous than downregulated genes. Nevertheless, both up- and downregulated genes that overlap between CdLS and RAD21-TEV show enhancer proximity and RAD21/CTCF binding. This suggests that there is a core set of upregulated genes that is upregulated when the function of NIPBL and/or cohesin is compromised.

9 -In Discussion

“In addition to directly cohesin-dependent genes, reduced cohesin function in CdLS neurons also affected genes that were not sensitive to acute cohesin depletion in mouse neurons. These lacked hallmarks of direct cohesin targets.” Cryptic. The concept of cohesin-target gene needs to be clarified. The fact is that most genes affected in CdLS are not affected in mouse neurons after Rad21 depletion. Also, there is no formal proof of “reduced cohesin function” in CdLS neurons in the current manuscript (see question 3 above).

We thank the referee for raising this point. We have revised the discussion to specify 'enhancer proximity and RAD21/CTCF binding' instead of hallmarks of direct cohesin targets. As detailed in our response to point 2 above, NIPBL mutations have been shown to reduce the association of cohesin with chromatin, both in CdLS patient cells (Liu et al., 2009, revised ref. 14) and in model systems (revised references 14-17), but we agree that we have no formal proof of reduced cohesin function in the CdLS neurons analysed in the current manuscript.

- The authors could consider (and even discuss) the possibility that some deregulated genes in CdLS patients are the consequence of a role of NIPBL independent of cohesin. Moreover, it is formally possible that (most, several, some) genes found deregulated in both CdLS patients carrying NIPBL mutations and Rad21-depleted mouse neurons are regulated by NIPBL and cohesin through independent mechanisms.

We fully agree with the referee. In response, we have extensively revised the narrative of our manuscript. The introduction now provides a more balanced account of current knowledge about the relationship between NIPBL and cohesin, taking care to cite key studies listed by the referee: "Heterozygous or hypomorphic germline mutations of cohesin and associated factors such as the cohesin loading factor NIPBL result in reduced cohesin function¹⁴⁻¹⁷. Although compatible with normal cell cycle progression, such mutations result in a human developmental disorder known as Cornelia de Lange Syndrome (CdLS)¹⁸⁻²⁰. All CdLS patients show a degree of intellectual disability, and 60 to 65% have autism spectrum disorder (ASD), often in the absence of structural brain abnormalities or neurodegeneration^{19,21,22}.

Experimental perturbations of neuronal cohesin and cohesin-related factors such as Nipbl during mouse development have shown changes in animal behaviour as well as abnormal neuronal morphology²³, migration²⁴ and gene expression²³⁻²⁶. NIPBL has historically been considered as separate from the core cohesin complex, and with distinct^{24,27,28} and sometimes even antagonistic activities²⁹. However, current structural³⁰ and functional³¹ evidence indicates that NIPBL is integral to the cohesin complex in its loading state³⁰ and contributes to ATPase activity³¹ during loop extrusion, the process thought to form chromatin loops, TADs and contact domains⁵. Accordingly, the depletion of NIPBL and cohesin have similar or identical effects on 3D genome organisation^{7,8}. Nevertheless, it remains unclear to what extent gene expression changes in cells with *Nipbl* mutations relate to cohesin versus potential other functions of NIPBL ref. 24,27,28)." .

As discussed above, the revised Results section presents new experimental data (RNA-seq of *Nipbl*^{+/-} mouse neurons) and analysis (analysis of new RNA-seq of *Nipbl*^{+/-} mouse neurons and of published microarray data for whole *Nipbl*^{+/-} mouse brain). The RNA-seq experiments in *Nipbl*^{+/-} mouse neurons enabled us to directly compare gene expression in 24h RAD21-depleted and *Nipbl*^{+/-} mouse neurons.

The discussion puts these results into context and we now avoid any claim to the effect that experimental cohesin depletion mimics all aspects of gene deregulation in CdLS.

Referee 3

We thank the referee for providing insightful and constructive comments, and the assessment that the 'study is interesting and demonstrates the utility of exploring the consequences of disease-causing genes/complexes outside the context of germline mutation or deletion'

Given the widespread disturbance of neuronal genes upon acute RAD21 cleavage, it would be important to understand if sustained RAD21 cleavage affected neuronal maturation and function. If sustained RAD21 depletion does not result in downstream neuronal dysfunction, then the functional impact of these acute gene expression changes is unclear, as we know the mouse model demonstrates these deficiencies. For example, the authors could measure neuronal morphology, synaptic marker changes at the protein level, or electrophysiology to show the gene expression changes have an impact on the neuronal biology. These experiments will be important in benchmarking this tool as a method to study cohesin, neuronal function, and pathogenesis of CdLS.

Does sustained RAD21 depletion (for example for 10-14 days) cause further gene expression changes?

We have characterised the impact of sustained RAD21-TEV cleavage on gene expression by new RNA-seq experiments at 1, 3 (Supplementary Fig. 8a, b, Supplementary Data 12) and 7 days (Supplementary Fig. 10, shown below in response to the referees' point 3, Supplementary Data 14) after RAD21-TEV cleavage.

We found a progressive increase in the number of deregulated genes.

Reduced mRNA expression translated into reduced levels of protein in RAD21-TEV neurons, as illustrated for NLGN1, encoded by *Nlgn1* (Supplementary Fig. 8c, d).

Supplementary Figure 8. RAD21-TEV depletion results alters the expression of protein as well as mRNA.

a) Western blot of RAD21-TEV protein expression at 24, 48 and 72 h of 4-OHT treatment.

b) Volcano plot of gene expression log₂ fold-change versus adjusted *P* value in RAD21-TEV neurons transduced with ERT2-TEV and treated with 4-OHT for 24h (n = 3, Supplementary Data 8) or 72h (n = 3, Supplementary Data 12). *Nlgn1* is deregulated (adj. *P* = 6.42 × 10⁻⁶, log₂ fold-change = -0.69 at 24h and adj. *P* = 0.42 × 10⁻⁶, log₂ fold-change = -0.69 at 72h) and is highlighted with a black circle, *Syn1* is not deregulated (adj. *P* = 0.92, log₂ fold-change = 0.06 at 24h and adj. *P* = 0.79, log₂ fold-change = -0.1 at 72h) and is highlighted in grey.

c) Western blot of deregulated NLGN1 protein expression 24 and 72 h after 4-OHT or EtOH treatment. Bar plot of NLGN1 protein expression normalised to LAMIN B (n = 2).

d) Western blot of non-deregulated control protein SYN1 following 24 and 72 h after 4-OHT or EtOH treatment. Bar plot of SYN1 protein expression normalised to LAMIN B (n = 2).

These conclusions are corroborated by results from a parallel study (Calderon, Weiss, et al., in preparation) where we used genetics to delete conditional *Rad21*^{lox} alleles specifically in immature post-mitotic cortical and hippocampal excitatory neurons using *Nex*^{Cre}. RNA-seq showed similar gene expression defects to RAD21-TEV (referee Fig. NexCre b), and highly significant numerical overlap with genes (*P* < 2.22e-16, odds ratio = 8.24). Explant cultures of wild-type and *Rad21*^{lox/lox} *Nex*^{Cre} E18.5/19.5 cortical neurons were maintained for 10 days in vitro to allow for neuronal maturation. Sparse labeling with GFP was used to visualize

processes of individual neurons, and Sholl analysis to quantitate the number of axonal crossings, the length of dendrites, the number of terminal points, the number of branch points and the number of spines in GAD67-negative *Rad21*^{+/+} *Nex*^{Cre} and GAD67-negative *Rad21*^{lox/lox} *Nex*^{Cre} neurons. Based on these measures, cohesin-deficient *Rad21*^{lox/lox} *Nex*^{Cre} neurons showed impaired morphological complexity (Referee Figure NexCre b,c). Although not part of the current manuscript, these data confirm the referees' suggestion that depletion of cohesin affects neuronal phenotype.

Referee Figure NexCre. Loss of cohesin from immature post-mitotic neurons perturbs neuronal maturation

a) Analysis of gene ontology of biological functions of deregulated genes in *Rad21*^{lox/lox} *Nex*^{Cre} neurons. Enrichment is calculated relative to expressed genes.

b) Morphology of *Rad21*^{+/+} *Nex*^{Cre} and *Rad21*^{lox/lox} *Nex*^{Cre} cortical neurons in explant culture on rat glia. Cultures were sparsely labeled with GFP to visualize individual cells and their processes, and stained for GAD67 to exclude GABAergic neurons. Dendritic traces of GFP⁺ neurons. Scale bar = 50 μm.

c) Shall analysis of *Rad21*^{+/+} *Nex*^{Cre} and *Rad21*^{lox/lox} *Nex*^{Cre} cortical neurons in explant cultures shown in E. Shown is the number of crossings, dendritic length, terminal points, branch points and spines per 10 μ m. Three independent experiments, 32 *Rad21*^{lox/lox} *Nex*^{Cre} and 28 *Rad21*^{+/+} *Nex*^{Cre} neurons except for the number of spines (two independent experiments, 10 *Rad21*^{lox/lox} *Nex*^{Cre} and 10 *Rad21*^{+/+} *Nex*^{Cre} neurons). * adj. *P* <0.05, ** adj. *P* <0.01, *** adj. *P* <0.001, **** adj. *P* <0.0001. Scale bar = 10 μ m.

3) Does the genomic architecture return to the wild-type state upon rescue of cohesin? Or does the architecture of the genome adapt to a new steady state that is somewhat different than an unperturbed neuron. If so, do these differences account for some of the gene expression changes that do not rescue or that occur de novo upon RAD21-depletion? The authors can perform chromosome conformation capture (ideally Hi-C) in the RAD21-rescue experiment to address this question.

We were both excited and concerned about the referees' idea that neurons might adapt to the cohesin-depleted state. If true, this would question the interpretation of the gene expression rescue, which forms a key part of the paper.

We have extended the analysis of neurons 24h after RAD21-TEV depletion by new RNA-seq data after continued cohesin depletion for 3 and 7 days (Supplementary Fig. 8, above, and Supplementary Fig. 10, below). These experiments provide an important test for the interpretation of the observed gene expression rescue by asking whether the observed reversibility of gene deregulation after transient RAD21-TEV cleavage requires the restoration of cohesin. The data show that neurons do not simply adapt to the loss of cohesin. In fact, gene expression changes became more extensive with extended time after the depletion cohesin (Supplementary Fig. 8, above, and Supplementary Fig. 10, below). These new experiments provide support for the conclusion that the observed reversal of gene expression indeed requires the restoration of cohesin.

Our attempts to carry out chromosome conformation capture in the RAD21-rescue system have so far been unsuccessful, partly due to disruption caused by COVID-19. However, we feel that 3D rescue experiments have perhaps become less of a priority for two reasons. First, our new RNA-seq data link rescue of gene expression to the restoration of cohesin, rather than to adaptation. Second, published work shows that 3D contacts are restored after RAD21 depletion and re-expression (Rao et al., 2017, ref. 7; Vian et al., 2018, <https://doi.org/10.1016/j.cell.2018.03.072>). With regard to the referee's interesting question whether 3D contacts can account for the few genes that escape rescue, we ask the referee

to consider that even with the highest resolution Hi-C maps it can be difficult to ascribe the deregulation of specific genes to subtle changes in 3D contacts.

Supplementary Figure 10. Neuronal gene expression does not accommodate to depletion of RAD21.

a) Western blot of RAD21-TEV protein expression 7 days hours after Dox pulse (24h, 1 μ g/ml). Bar plot of RAD21-TEV protein expression normalised to LAMIN B 7 days after Dox exposure, ~20% RAD21-TEV protein remained (n = 3, d = days, error bars = range).

b) Volcano plot of gene expression \log_2 fold-change versus adjusted P value in RAD21-TEV +Dox (7 days hours after 24-hour 1 μ g/ml pulse, n = 3). 2836 genes were down- and 1920 genes were upregulated (adj $P < 0.05$, shown in red, DE = differentially expressed, Supplementary Data 14).

REVIEWER COMMENTS

Reviewer #1 (Remarks to the Author):

In this revised version of the manuscript by Weiss et al, authors have addressed a number of experimental issues and the manuscript is now improved.

Experimentally, I would have liked to see what happens if they analyze transcriptome on the context of the RAD21-TEV allele in heterozygosis, or alternatively trying to shorten cleavage time of RAD21 in order to have milder depletion conditions, as I suggested in my comments.

In this regard, I am also disappointed that authors are not addressing one of my last comments in relation to some papers not cited or not properly discussed:

- Previous reports show that RAD21 depletion has no major consequences on general transcription, excepting some genes dependent on distal elements as indicated by authors. Some of these papers are not cited or discussed (Fisher et al (2017) *Leukemia* 31: 712, not cited; Schwarzer (2017) *Nature* 551: 51, cited in the introduction but not discussed). Similarly, some recent studies with neurodevelopment-related transcriptomic results in the context of CdLS are not cited or discussed (Luna-Pelaez et al (2019) *Cell Death Dis* 19:548, not cited; van den Berg (2017) *Neuron* 93: 348, cited in the introduction but not discussed).

- Finally, I am still not convinced on the title.

Reviewer #2 (Remarks to the Author):

The manuscript has improved and the authors have addressed many of my questions. However, one important one (below) has remained unanswered. While this work supports the idea that at least part of the gene deregulation observed in neurons from CdLS patients may be a consequence of impaired cohesin function, it remains unclear how reducing (by half?) functional NIPBL levels in the cell affects cohesin.

2- How do the NIPBL mutations identified in patients affect cohesin's presence on chromatin in general and at particular genome locations?

Here I was requesting the authors to show how a heterozygous mutation in NIPBL affects cohesin loading and distribution. The revised manuscript includes RNA-seq data from mNIPBL heterozygous neurons, and the same cells could be used to check cohesin distribution by ChIPseq and even to compare 4C seq maps obtained in these NIPBL heterozygous cells with those in RAD21-TEV. In this way, one could assess how a reduction in NIPBL levels affects cohesin's function in facilitating enhancer-promoter contacts, as proposed in Discussion (page 20). The authors show in figure 2h how complete cohesin depletion in neurons changes chromatin contacts between enhancers and promoters at the Protocadherin locus and these changes correlate with downregulation of PCDHB10, which is also observed in neurons from CdLS patients. However, the same gene is not significantly downregulated in the murine NIPBL heterozygous neurons, suggesting that in this case the mechanism of downregulation

may be different in cohesin-depleted and CdLS neurons. How many of the fifty something genes that are deregulated in CdLS and RAD21 deficient neurons are also deregulated in neurons from NIPBL heterozygous embryos?

Also, I did not mention this in my first review, but I think the title is misleading and should be changed to something like

“Re-expression of cohesin following acute depletion in neurons restores the expression of CdLS-related genes”

Minor

Page 5, “RNAseq identified.... even though NIPBL RNA levels were not significantly altered by the NIPBL loss of function mutations in these patients (Supplementary Fig. 3b).”

The authors removed these data from current version but forgot to modify the main text accordingly.

The 4C seq analyses in Fig2 h, the Figure legend indicates n=4. Is the image shown the result of merging data from all 4 replicates or does it correspond to one of them? How similar are these replicates? The 4Cseq data have not uploaded in GEO and there is no information about it (total and mapped reads, for instance).

Reviewer #3 (Remarks to the Author):

The authors have done an excellent job addressing my comments and improving the manuscript through the additional data and expansion of the text.

Response to referees' comments

We thank the referees for their input on our revised manuscript, and were pleased to hear that the first round of revisions had substantially improved the paper. This said, we acknowledge that referees 1 and 2 had remaining questions, specifically about (i) the impact of cohesin on gene expression, (ii) the relationship between cohesin and NIPBL, and (iii) the extent of similarity of gene expression in response to the perturbation of cohesin versus NIPBL. We will address these questions in detail in our point-by-point response below.

In revising the manuscript a second time, we now provide a specific account of cohesin's role in gene expression, in particular the expression of enhancer-associated genes, inducible genes, and neuronal activity-dependent genes.

We have added a detailed section in the role of NIPBL in cohesin loading, and the impact of heterozygous *Nipbl* mutations on global and locus-specific cohesin binding and 3D chromatin contacts.

We have made specific reference to the possibility that NIPBL may serve additional, cohesin-independent functions in transcription, and that these may be unrelated to its role in cohesin loading. The resulting changes include the citation of papers requested by referee 1.

Finally, we have changed the title as requested by referees 1 and 2.

The resulting changes are marked in red in the revised manuscript.

We hope that the referees and editors will find that these revisions have further improved the paper.

Reviewer comments - our point-by-point response

Reviewer #1 (Remarks to the Author):

In this revised version of the manuscript by Weiss et al, authors have addressed a number of experimental issues and the manuscript is now improved.

We thank the referee for the encouraging comment that the manuscript is now improved.

Experimentally, I would have liked to see what happens if they analyze transcriptome on the context of the RAD21-TEV allele in heterozygosis, or alternatively trying to shorten cleavage time of RAD21 in order to have milder depletion conditions, as I suggested in my comments.

The referee raises an important point, which we have addressed this by comparing moderate and strong depletion of RAD21-TEV in 2 different depletion systems. We apologize if we did not communicate this point with sufficient clarity in our response to the

referee. We try to rectify this here. The two RAD21-TEV cleavage systems presented in our study differ with respect to the depth of cohesin depletion: ERT2-TEV depleted RAD21-TEV by ~85% within 8h of 4-OHT addition (Fig. 2c), while NLS-TEV induction resulted in a milder depletion with 30% residual RAD21-TEV 24h after Dox addition (Supplementary Fig. 6c). As shown in Fig. 2g, both depletion systems resulted in highly similar changes in gene expression ($P < 2.2 \times 10^{-16}$). These data show that the precise depth of cohesin depletion is not critical for the observed impact on gene expression.

Figure 2g. Scatter plot of gene expression, comparing \log_2 fold-change of deregulated genes (adj $P < 0.05$) in response to acute cohesin depletion induced by ERT2-TEV and NLS-TEV (DE = differentially expressed, R = Pearson correlation coefficient).

In addition, we have shown strong overlap between changes in gene expression in RAD21-TEV and heterozygosity in *Nipbl*, both by analysing published data from *Nipbl*^{+/-} embryonic mouse brain and by generating new RNA-seq data from *Nipbl*^{+/-} neurons (Supplementary Fig. 7e).

Supplementary Fig. 7e. GSEA showing the expression of in RAD21-TEV neurons of genes downregulated *Nipbl*^{+/-} neurons (DEseq2, adj. $P < 0.05$) and in embryonic *Nipbl*^{+/-} brain²³ (DEseq2, adj. $P < 0.05$).

In this regard, I am also disappointed that authors are not addressing one of my last comments in relation to some papers not cited or not properly discussed:

- Previous reports show that RAD21 depletion has no major consequences on general transcription, excepting some genes dependent on distal elements as indicated by authors.

The impact of cohesin on transcription is an important question, which has been the focus of work in our lab for over a decade (Seitan et al., 2011 Nature ; Seitan et al., Genome Research 2013, Ing-Simmons et al., Genome Research 2015; Cuartero, Weiss et al. Nat Immunol 19: 932-41, 2018). Although the global impact of cohesin depletion on gene expression is moderate with approximately 10% of genes found deregulated, these deregulated genes are strongly enriched for genes that are dynamically expressed, including cell type-specific and inducible genes. In fact, the transcriptional response to microbial signals was severely disrupted in cohesin-deficient macrophages (Cuartero, Weiss et al. Nat Immunol 19: 932-41, 2018). We have explained this more clearly in the introduction to the revised manuscript: ' **While the loss of cohesin affects the transcription of a limited number of genes, enhancer-associated^{7,13} and inducible genes¹⁴, including neuronal activity-dependent genes¹⁵, are frequently deregulated when 3D organisation is perturbed by the loss of cohesin.** '. In the current paper we show that the expression of cohesin-dependent genes is largely rescued by the restoration of cohesin.

Some of these papers are not cited or discussed (Fisher et al (2017) Leukemia 31: 712, not cited; Schwarzer (2017) Nature 551: 51, cited in the introduction but not discussed). Similarly, some recent studies with neurodevelopment-related transcriptomic results in the context of CdLS are not cited or discussed (Luna-Pelaez et al (2019) Cell Death Dis 19:548, not cited; van den Berg (2017) Neuron 93: 348, cited in the introduction but not discussed).

The Fisher et al 2017 study is about the role of cohesin in the self-renewal of haematopoietic progenitors, and was appropriately cited in our previous study on cohesin, inducible gene expression and acute myeloid leukaemia, where this reference was most relevant (Cuartero, Weiss et al. Nat Immunol 19: 932-41, 2018).

In the revised manuscript we now cite Luna-Pelaez et al (2019, new reference 27) in the context of potential cohesin-independent functions of NIPBL (interaction with BRD4).

The main message of Schwarzer et al is that the formation of TADs, but not of A/B compartments requires Nipbl/cohesin, and the paper is cited for this finding in our current study.

As requested by the referee, we have revised our discussion, which now directly refers to van den Berg et al, 2017, Zuin et al, 2014, Nolen et al, 2013, and Luna-Pelaez et al, 2019. These revisions include the statement that '**in addition to directly cohesin-dependent genes, CdLS neurons also showed deregulation of genes that lacked hallmarks of direct cohesin targets and were not sensitive to acute cohesin depletion in mouse neurons. This may be due to cohesin-independent functions of NIPBL (ref. 22,25-27) or to secondary effects of reduced cohesin function on the expression of other genes**'.

- Finally, I am still not convinced on the title.

We have changed the title to '**Neuronal genes deregulated in Cornelia de Lange Syndrome are partially rescued by cohesin**', which more accurately reflects the findings of our study

Reviewer #2 (Remarks to the Author):

We are pleased to hear that the referee found the manuscript improved and that many of the questions raised had been addressed.

We also note that the referee was satisfied with the conclusion that at least part of the gene deregulation observed in neurons from CdLS patients may be a consequence of impaired cohesin function.

However, one important one (below) has remained unanswered. While this work supports the idea that at least part of the gene deregulation observed in neurons from CdLS patients may be a consequence of impaired cohesin function, it remains unclear how reducing (by half?) functional NIPBL levels in the cell affects cohesin. How do the NIPBL mutations identified in patients affect cohesin's presence on chromatin in general and at particular genome locations? Here I was requesting the authors to show how a heterozygous mutation in NIPBL affects cohesin loading and distribution. The revised manuscript includes RNA-seq data from mNIPBL heterozygous neurons, and the same cells could be used to check cohesin distribution by ChIPseq and even to compare 4C seq maps obtained in these NIPBL heterozygous cells with those in RAD21-TEV. In this way, one could assess how a reduction in NIPBL levels affects cohesin's function in facilitating enhancer-promoter contacts, as proposed in Discussion (page 20).

We thank the referee for raising this important point. The question of how reducing functional NIPBL levels in the cell affects cohesin has been addressed extensively in the literature, as published both by authors of the current manuscript and by others. Although we cited the relevant papers in the previous version of our manuscript, we apologise if we did not make this point clearly enough.

Specifically, the Krantz lab investigated cohesin binding to chromatin and gene expression in lymphoblastoid cells from CdLS patients with *NIPBL* mutations. ChIP array showed a 29.7% reduction in the number of cohesin binding sites (9,350 vs. 13,560) in CdLS (Liu, J. et al. Transcriptional Dysregulation in NIPBL and Cohesin Mutant Human Cells. *PLoS Biology* 7, e1000119 (2009), reference 31).

Newkirk et al (2017) showed that *Nipbl*^{+/-} mouse embryonic fibroblasts had reduced numbers of RAD21 binding sites as determined by ChIP-seq for the cohesin subunit RAD21. 3D chromatin contacts were reduced, and cohesin-bound genes were disproportionately downregulated in *Nipbl*^{+/-} cells (Newkirk et al. The effect of Nipped-B-like (Nipbl) haploinsufficiency on genome-wide cohesin binding and target gene expression: modeling Cornelia de Lange syndrome. *Clinical Epigenetics* 9, (2017), reference 32).

To illustrate this point, Referee Fig. 1 shows RAD21 ChIP-seq and control tracks at 4 relevant loci; *Cebpa*, which is the focus of the original study by Newkirk et al (2017), the

prototypic cohesin target gene *Myc*, and *Lpar1* and *Vip*, two genes found downregulated in both *Nipbl*^{+/-} neurons and *Nipbl*^{+/-} MEFs (*Lpar1*, Kawauchi et al., 2009; *Vip*, Newkirk et al., 2017).

Referee Figure 1. Cohesin binding as assessed by RAD21 ChIP-seq in control and *Nipbl*^{+/-} mouse embryonic fibroblasts. RAD21 ChIP-seq and control ChIP tracks for 4 genes. *Lpar1* and *Vip* are downregulated in *Nipbl*^{+/-} neurons and *Nipbl*^{+/-} MEFs (Kawauchi et al., 2009 reference 23; Newkirk et al., 2017 reference 32)

Chien et al (2011) found reduced cohesin binding and reduced 3D chromatin contacts at the β -globin locus in *Nipbl*^{+/-} fetal liver cells (Chien, R. et al. Cohesin Mediates Chromatin Interactions That Regulate Mammalian β -globin Expression. *Journal of Biological Chemistry* 286, 17870-17878 (2011), reference 33).

Perhaps most importantly in the context of the current study, Remeseiro et al (2013) found that cohesin/SMC1 binding was reduced in *Nipbl*^{+/-} embryonic mouse brain, specifically at genes that were deregulated (Remeseiro et al. Reduction of *Nipbl* impairs cohesin loading locally and affects transcription but not cohesion-dependent functions in a mouse model of Cornelia de Lange Syndrome. *Biochimica et Biophysica Acta (BBA) - Molecular Basis of*

Disease 1832, 2097-2102 (2013), reference 34). This analysis included *Pcdhb* genes and therefore directly relates to findings of our current study.

Referee Fig 2 shows a analysis of cohesin binding at deregulated Protocadherin genes in *Nipbl*^{+/-} embryonic mouse brain. This analysis is based on original data by Remeseiro et al., 2013 (reference 34), which were kindly provided by Prof Ana Losada.

Referee Fig 2. Reduced cohesin binding at deregulated *Pcdh* genes in *Nipbl*^{+/-} embryonic mouse brain. ChIP-qPCR for the cohesin subunit SMC1A in wild-type (*Nipbl*^{+/+}) and *Nipbl* heterozygous (*Nipbl*^{+/-}) mouse brain at embryonic day 17.5 (e17.5) at the promoters of 5 downregulated *Pcdh* genes (*Pcdh7*, *Pcdh17*, *Pcdh11x*, *Pcdhb17*, and *Pcdhb21*; Remeseiro et al., 2013). Mean and SE of ChIP-PCR signals in *Nipbl*^{+/-} brain are shown normalized to wild-type. Each data point represents one *Pcdh* gene promoter.

Both Newkirk et al (2017) and Chien et al. (2011) showed that reduced cohesin binding was accompanied by defective 3D chromatin contacts in *Nipbl*^{+/-} mouse embryonic fibroblasts and fetal liver cells (references 32, 33), providing a candidate mechanism for "how a reduction in NIPBL levels affects cohesin's function in facilitating enhancer-promoter contacts, as proposed in Discussion (page 20)".

To communicate these points more clearly in our manuscript, we have revised our introduction to state: '**Consistent with the function of NIPBL as a loading factor for cohesin, human *NIPBL*^{+/-} lymphoblastoid cells³¹, *Nipbl*^{+/-} mouse embryonic fibroblasts³² and fetal liver cells³³ show reduced global or local cohesin binding³¹⁻³³ and defective 3D chromatin contacts^{32,33} and reduced cohesin binding was found at deregulated genes in *Nipbl*^{+/-} embryonic mouse brain³⁴.**'

The authors show in figure 2h how complete cohesin depletion in neurons changes chromatin contacts between enhancers and promoters at the Protocadherin locus and these changes correlate with downregulation of PCDHB10, which is also observed in neurons from CdLS patients. However, the same gene is not significantly downregulated in the murine NIPBL heterozygous neurons, suggesting that in this case the mechanism of downregulation may be different in cohesin-depleted and CdLS neurons.

We thank the referee for this question, and the opportunity to explain the deregulation of genes in neurons from CdLS patients and mouse neurons heterozygous for *Nipbl* or

depleted of RAD21 by proteolytic cleavage. Referee Fig. 3 shows the expression of *Pcdhb* genes in these settings. The axes show log₂ fold-changes in each setting compared to control (normal human neurons for CdLS, *Nipbl*^{+/+} for *Nipbl*^{+/-}, and RAD21-TEV uncleaved for RAD21-TEV cleaved). Negative log₂ fold-changes indicate downregulation, positive log₂ fold-changes indicate upregulation, zero means no change. All *Pcdhb* genes show the same trend, except perhaps *PCDHB8* in CdLS, where it is neither up- nor downregulated. *P*-values show the probability of finding the observed downregulation of *Pcdhb* gene expression by chance and range from *P*=1.62e-07 for CdLS neurons to *P*=2.75e-13 (RAD21-TEV NLS).

PCDHB10, which was queried by the referee, is highlighted in red. The *PCDHB10* gene is downregulated to a similar extent in response to reduced expression of NIPBL or cohesin in all experimental settings we have examined..

Referee Figure 3. The expression of *Pcdhb* genes is cohesin- and NIPBL-dependent.

The expression of genes in the *Pcdhb* cluster is shown for CdLS neurons, mouse neurons heterozygous for *Nipbl*, and mouse neurons 24h after RAD21-TEV cleavage by ERT2-TEV and NLS-TEV. *Pcdhb10* is highlighted in red. *P*-values were determined by Wilcoxon gene set test and show the probability of finding the observed changes in *Pcdhb* gene expression by chance.

These data make the point that whether a particular gene meets the criteria for statistically significant deregulation reflects factors such as variability between replicates, the number of genes analysed in the same experiment (which determines the penalty for multiple testing - high in the case of genome-wide analyses), etc. As illustrated by the *Pcdhb* gene family, coherence between systems is evident from correlations in the direction of gene expression

changes and gene set enrichment analysis (illustrated in the manuscript in Fig. 2g, Fig. 3a, and Supplementary Fig. 7e, and below as Referee Fig. 4).

How many of the fifty something genes that are deregulated in CdLS and RAD21 deficient neurons are also deregulated in neurons from NIPBL heterozygous embryos?

The numerical overlap is 14. However, as illustrated above for *Pcdhb* genes, numerical overlap between genes deregulated across experimental settings is just one facet of deregulated gene expression in response to loss of cohesin or NIPBL mutations. Coherence between systems is evident from the correlated changes in the direction of gene expression. This coherence is illustrated in Fig. 2g, Fig. 3a, and Supplementary Fig. 7e.

Fig. 2g shows that the expression of individual genes shows coherent changes in direction between the two RAD21-TEV depletion systems, ERT2-TEV and NLS-TEV. Regardless of whether the expression change is statistically significant for particular genes, their expression is overwhelmingly changed in the same direction in both systems. The observed *P*-value of $< 2.2 \times 10^{-16}$ indicates that this coherence is unlikely to be observed by chance

Figure 2g. Scatter plot of gene expression, comparing \log_2 fold-change of deregulated genes ($\text{adj } P < 0.05$) in response to acute cohesin depletion induced by ERT2-TEV and NLS-TEV (DE = differentially expressed, *R* = Pearson correlation coefficient).

Fig. 3a illustrates this coherence for gene expression changes between CdLS and RAD21-TEV: here, all genes that are significantly downregulated 24h after RAD-TEV depletion are plotted in CdLS, and the great majority of these genes show downregulation also in CdLS (left). Conversely, the great majority of genes that are significantly downregulated in CdLS also show downregulation 24h after RAD-TEV depletion (right, false discovery rate = 0).

Figure 3a. GSEA of RAD21-TEV downregulated genes in CdLS NeuN-positive RNAseq (left). GSEA of CdLS NeuN-positive downregulated genes in RAD21-TEV (right). NES = normalised enrichment score, FDR = false discovery rate.

Supplementary Fig. 7e illustrates this coherence for gene expression changes between RAD21-TEV and *Nipbl*^{+/-} neurons as well as embryonic *Nipbl*^{+/-} brain. Specifically, genes that are downregulated in *Nipbl*^{+/-} neurons (left) or embryonic *Nipbl*^{+/-} brain (right) are also downregulated 24h after RAD-TEV depletion. The false discovery rate is 0, emphasizing that the coherence in gene deregulation exceeds the numerical overlap of statistically significant gene expression changes.

Supplementary Fig. 7e. GSEA showing the expression of in RAD21-TEV neurons of genes downregulated *Nipbl*^{+/-} neurons (DEseq2, adj. *P* < 0.05) and in embryonic *Nipbl*^{+/-} brain²⁵ (DEseq2, adj. *P* < 0.05).

Similar, albeit somewhat weaker correlations are found in the expression of genes in adult human CdLS neurons *Nipbl*^{+/-} mouse neurons (referee Fig. 4).

Referee Figure 4. Coherence of gene expression between CdLS and *Nipbl*^{+/-} mouse neurons. Gene set enrichment analysis for the expression of genes in adult human CdLS neurons, embryonic *Nipbl*^{+/-} mouse brain and embryo-derived mouse *Nipbl*^{+/-} neurons.

Also, I did not mention this in my first review, but I think the title is misleading and should be changed to something like "Re-expression of cohesin following acute depletion in neurons restores the expression of CdLS-related genes"

We have changed the title to '**Neuronal genes deregulated in Cornelia de Lange Syndrome are partially rescued by cohesin**', which more accurately reflects the findings of our study

Minor

Page 5, "RNAseq identified.... even though NIPBL RNA levels were not significantly altered by the NIPBL loss of function mutations in these patients (Supplementary Fig. 3b)."

The authors removed these data from current version but forgot to modify the main text accordingly.

We thank the referee for spotting this error and have revised the manuscript accordingly.

The 4C seq analyses in Fig2 h, the Figure legend indicates n=4. Is the image shown the result of merging data from all 4 replicates or does it correspond to one of them? How similar are these replicates? The 4Cseq data have not uploaded in GEO and there is no information about it (total and mapped reads, for instance).

We thank the referee for noting this omission. We are adding the 4C-seq data to the GEO submission. We have modified the legend for Fig. 2h legend to state: 'The colour panel shows the mean contact intensities for multiple window sizes from 2kb to 5kb for n = 4 independent biological replicates. All replicates showed comparable results and were merged to generate this figure.' The individual replicates are shown in Referee Fig. 5.

Referee Fig. 5. Individual biological replicates and merged data for the 4C analysis of the *Pcghb* locus. Relates to Fig. 2h.

Reviewer #3 (Remarks to the Author):

The authors have done an excellent job addressing my comments and improving the manuscript through the additional data and expansion of the text.

We thank the referee for these encouraging comments.

REVIEWERS' COMMENTS

Reviewer #1 (Remarks to the Author):

I appreciate the effort made by the authors in revising the manuscript again. Now they have answered to all my experimental concerns. I just want to mention that the new title is not the best to reflect the real contribution of this work. It suggests that expression of cohesin could substantially solve the deficiencies associated with CdLS, while in most CdLS patients (those with mutations in NIPBL) this certainly would not be the case. I would have preferred a title in the line suggested by Reviewer 2. However, the manuscript is of high interest and provides relevant new data, so I consider it now appropriate for publication in *Natura Communications*.

Reviewer #2 (Remarks to the Author):

I thank the authors for their efforts and their explanations to my queries. I support the publication of this version of the manuscript.

Response to referees

We thank the referees for their time and effort in assessing our manuscript

We suggest the title '**Neuronal genes deregulated in Cornelia de Lange Syndrome respond to removal and re-expression of cohesin**' to meet the remaining concern voiced by referee 1.